# DMRT1-mediated reprogramming drives development of cancer resembling human germ cell tumors with features of totipotency

Jumpei Taguchi[1], Hirofumi Shibata[2,10], Mio Kabata[2], Masaki Kato[3,11], Kei Fukuda [3], Akito Tanaka[2], Sho Ohta[1], Tomoyo Ukai[1], Kanae Mitsunaga[2], Yosuke Yamada[2,12], So I Nagaoka[4,13], Sho Yamazawa[5], Kotaro Ohnishi[2,14], Knut Woltjen [2], Tetsuo Ushiku [5], Manabu Ozawa[6], Mitinori Saitou[2,4,7], Yoichi Shinkai [3], Takuya Yamamoto [2,7,8,9] & Yasuhiro Yamada [1,9 ✉]

In vivo reprogramming provokes a wide range of cell fate conversion. Here, we discover that in vivo induction of higher levels of OSKM in mouse somatic cells leads to increased expression of primordial germ cell (PGC)-related genes and provokes genome-wide erasure of genomic imprinting, which takes place exclusively in PGCs. Moreover, the in vivo OSKM reprogramming results in development of cancer that resembles human germ cell tumors. Like a subgroup of germ cell tumors, propagated tumor cells can differentiate into tropho-blasts. Moreover, these tumor cells give rise to induced pluripotent stem cells (iPSCs) with expanded differentiation potential into trophoblasts. Remarkably, the tumor-derived iPSCs are able to contribute to non-neoplastic somatic cells in adult mice. Mechanistically, DMRT1, which is expressed in PGCs, drives the reprogramming and propagation of the tumor cells in vivo. Furthermore, the DMRT1-related epigenetic landscape is associated with trophoblast competence of the reprogrammed cells and provides a therapeutic target for germ cell tumors. These results reveal an unappreciated route for somatic cell reprogramming and underscore the impact of reprogramming in development of germ cell tumors.

[1] Division of Stem Cell Pathology, Center for Experimental Medicine and Systems Biology, Institute of Medical Science, The University of Tokyo, Minoto-ku, Tokyo, Japan. [2] Department of Life Science Frontiers, Center for iPS Cell Research and Application (CiRA), Kyoto University, Sakyo-ku, Kyoto, Japan. [3] Cellular Memory Laboratory, RIKEN Cluster for Pioneering Research, Wako-shi, Saitama, Japan. [4] Department of Anatomy and Cell Biology, Graduate School of Medicine, Kyoto University, Yoshida-Konoe-cho, Sakyo-ku, Kyoto, Japan. [5] Department of Pathology, Graduate School of Medicine, The University of Tokyo, Tokyo, Japan. [6] Laboratory of Reproductive Systems Biology, Center for Experimental Medicine and Systems Biology, Institute of Medical Science, The University of Tokyo, Tokyo, Japan. [7] Institute for the Advanced Study of Human Biology (WPI-ASHBi), Kyoto University, Yoshida-Konoe-cho, Sakyo-ku, Kyoto, Japan. [8] Medical-risk Avoidance Based on iPS Cells Team, RIKEN Center for Advanced Intelligence Project (AIP), Kyoto, Japan. [9] AMED-CREST, AMED 1-7-1 Otemachi, Chiyodaku, Tokyo, Japan. [10] Present address: Department of Otolaryngology, Gifu University Graduate School of Medicine, Gifu, Japan. [11] Present address: Laboratory for Transcriptome Technology, RIKEN Center for Integrative Medical Sciences, Yokohama, Japan. [12] Present address: Department of Diagnostic Pathology, Kyoto University Hospital, Kyoto, Japan. [13] Present address: Department of Embryology, Nara Medical University, Nara, Japan. [14] Present address: Department of Gastroenterology/Internal Medicine, Gifu University Graduate School of Medicine, Gifu, Japan. ✉email: yasu@ims.u-tokyo.ac.jp

Programming of somatic cells into pluripotent stem cells (PSCs) is a multistep process induced by the forced expression of reprogramming factors: OCT4 (encoded by *Pou5f1*, also known as OCT3 and OCT3/4), SOX2, KLF4, and c-MYC (OSKM)[1,2]. A stepwise reorganization of transcriptional and epigenetic regulation takes place during successful OSKM reprogramming. Notably, the details of the reprogramming process depend on cell-intrinsic factors such as cell type, as well as the levels and stoichiometries of reprogramming factors[3–7]. Previous studies also revealed that transient expression of OSKM in somatic cells leads to conversion into a variety of cell types, including neuronal cells and cardiomyocytes, in the presence of lineage-specifying extracellular cues in vitro[8,9], indicating that the extracellular environment guides the fate of OSKM-induced cells into different cell types. Taken together, these observations show that both cell-intrinsic and -extrinsic factors affect the process and destination of OSKM-induced reprogramming. To date, OSKM-mediated reprogramming processes have mostly been investigated at the cellular level in vitro. However, somatic cells are also reprogrammable in multicellular organisms in vivo[10–13]. Intriguingly, in vivo OSKM reprogramming leads to diverse biological responses, such as cancer development, senescence, and tissue regeneration[14,15], suggesting that the in vivo environment provides extrinsic signals that promote divergent cell-fate conversion in OSKM-induced cells.

Totipotency is defined as the ability of a single cell to divide and produce all the differentiated cells in an organism, including extra-embryonic tissues[16,17]. During early mammalian development, the immediate descendants of the zygotes differentiate into either the embryonic or extraembryonic lineage, and developmental potential is gradually restricted within each lineage after specification. Consequently, mouse embryonic lineage cells, including inner cell mass (ICM) cells and PSCs (the in vitro counterpart of ICM cells) basically lack the potential to differentiate toward an extraembryonic lineage. During the mammalian life cycle, embryonic lineage cells can become competent to undergo extraembryonic differentiation only through reprogramming during germ cell development and early pre-implantation embryogenesis following fertilization[18,19].

Although the differentiation potential of embryonic lineage cells is normally restricted within the lineage, a subgroup of germ cell tumors (GCTs), such as choriocarcinomas and a subset of embryonal carcinomas, contain extraembryonic cells even in adult organs comprising only embryonic lineage cells[20,21]. In addition, recent studies demonstrated that expansion of differentiation potential in PSCs (i.e., acquisition of the ability to differentiate into extraembryonic lineages) can be elicited by chemical inhibition of multiple signaling pathways in PSCs[22,23]. Similarly, induced PSCs (iPSCs) generated from adult somatic cells in vivo acquire totipotent cell-like differentiation potential[10]. Together, these results demonstrate that the potential to differentiate into an extraembryonic lineage can be induced in embryonic lineages under certain conditions. However, it remains unclear how embryonic lineage cells acquire the expanded differentiation potential into extraembryonic lineage cells.

In this study, we investigated the effects of various levels of OSKM expression and the in vivo environment on cellular reprogramming in adult mice. The results of this analysis revealed an unappreciated route for iPSC reprogramming. Our findings have important implications for GCT development and the acquisition of totipotency-like features by somatic cells.

## Results

### Generation of the D-OSKM reprogramming system.
To investigate the effects of varying levels of OSKM expression on cellular reprogramming, we first generated ESC lines carrying single or double doxycycline (Dox)-inducible OSKM polycistronic alleles (S-OSKM or D-OSKM) followed by an *ires-mCherry* cassette to visualize *OSKM* expression (Fig. 1a and Supplementary Fig. 1a)[11]. We then established mouse embryonic fibroblasts (MEFs) from animals derived from these ESCs. Upon Dox treatment, D-OSKM MEFs expressed higher levels of each reprogramming factor at both the mRNA and protein levels than S-OSKM MEFs (Fig. 1b, c). In addition, D-OSKM MEFs treated with a lower dose of Dox expressed lower levels of *OSKM* (Fig. 1b), demonstrating that OSKM levels were controllable by the number of reprogrammable alleles (D-OSKM or S-OSKM) and the concentration of Dox.

We next examined the effect of higher levels of OSKM on in vitro iPSC derivation from MEFs. Relative to S-OSKM alleles, D-OSKM alleles significantly increased the efficiency of NANOG-positive iPSC colony formation (Fig. 1d and Supplementary Fig. 1b). RNA sequencing (RNA-seq) of mCherry-positive D-OSKM/S-OSKM MEFs at various time points revealed that D-OSKM MEFs had transcriptional profiles distinct from those of S-OSKM MEFs following Dox exposure (Fig. 1e and Supplementary Fig. 1c,d). The results indicate that different levels of OSKM expression resulted in a difference in global transcriptional signatures, which presumably contributed to the differences in the efficiency of in vitro iPSC derivation following withdrawal of OSKM expression.

### In vivo induction of OSKM with S-OSKM/D-OSKM alleles.
The in vivo environment evokes a variety of cellular responses in the context of OSKM-induced reprogramming[14]. Therefore, we next investigated the effects of in vivo environments on somatic cell reprogramming with different levels of OSKM expression. For this purpose, we generated chimeric mice harboring S-OSKM/D-OSKM alleles by injecting S-OSKM/D-OSKM ESCs into blastocysts. Because D-OSKM chimeric mice became highly morbid soon after Dox treatment, we sought to induce *OSKM* in a tissue-specific manner by introducing a *loxP-stop-loxP* (*LSL*)-*rtTA* allele together with a tissue-specific *Cre* allele into the D-OSKM mouse model (Fig. 1f). The combination of the *LSL-rtTA* allele with the *Pax8-* and *Pdx1-Cre* allele allowed for specific induction of OSKM in the kidney and pancreas, respectively, where overt phenotypes had been detected during in vivo reprogramming[11]. As expected, we observed higher signal intensities of mCherry in the kidney cells of D-OSKM mice than in those of S-OSKM mice upon Dox treatment (Supplementary Fig. 1e,f). Indeed, D-OSKM–expressing kidney cells exhibited transcriptional profiles distinct from those of S-OSKM cells (Supplementary Fig. 1g).

### Development of D-OSKM tumors that resemble human germ cell tumors.
After 14-day induction of OSKM followed by withdrawal for seven days, we observed tumor development in the kidney and pancreas of both S-OSKM- and D-OSKM-reprogrammable mice (tumor incidence; S-OSKM mice [kidney: 16/20; pancreas: 8/20], D-OSKM mice [kidney: 17/24; pancreas: 18/21]) (Fig. 1g and Supplementary Fig. 2a). As reported previously[11], most S-OSKM tumors were teratomas composed of derivatives of all three germ layers, indicating that these tumors contain iPSCs (Fig. 1h and Supplementary Fig. 2b,c). By contrast, D-OSKM tumors lacked a mature teratoma component and exhibited propagation of OCT4-expressing poorly differentiated cells in both the kidney and pancreas (Fig. 1h, i and Supplementary Fig. 2b,d–f). Moreover, D-OSKM tumors exhibited invasive growth into the surrounding tissues and focal tumor necrosis, histological features of malignant tumors (Supplementary Fig. 2g). Indeed, D-OSKM pancreatic tumors occasionally invaded into the spleen (3/18 tumors) (Fig. 1j).

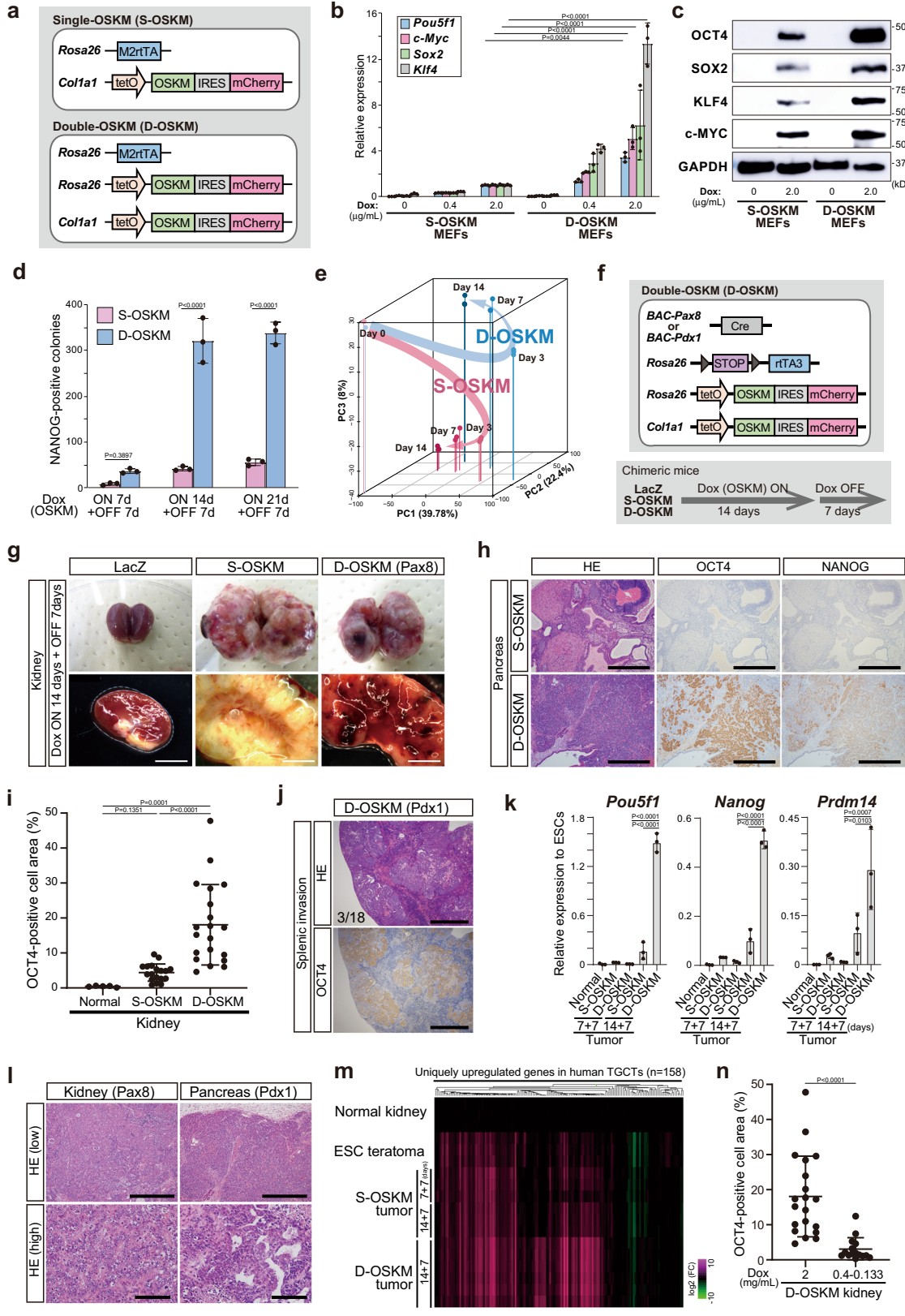

We found that OCT4-positive proliferating tumor cells simultaneously express NANOG (Fig. 1h, k and Supplementary Fig. 2b). Expression of these pluripotency markers is known as characteristics of several types of human GCTs such as embryonal carcinomas and seminomas[24,25]. Consistent with this, D-OSKM tumor cells arranged in solid sheets and in a glandular pattern,

had large, pleomorphic vesicular nuclei displaying prominent macronucleoli, poorly defined cytoplasmic membranes, and marked pleomorphism, which resembled the histology of human embryonal carcinomas[26] (Fig. 1l and Supplementary Fig. 2h). After inoculation of the minced D-OSKM tumor tissues, mice developed secondary tumors consisting of poorly differentiated

**Fig. 1 Development of D-OSKM tumors that resemble human germ cell tumors. a** Schematic illustration of the genomic construct of ESCs containing single- or double-OSKM polycistronic cassettes (S-/D-OSKM). **b** qRT-PCR analyses for expression of reprogramming factors in S-/D-OSKM MEFs at Day 3 after Dox treatment. Data are presented as means ± SD of biological triplicates. The mean expression level of S-OSKM MEFs treated with 2.0 μg/mL Dox was defined as 1 (two-way ANOVA and Tukey's multiple-comparison test, two-sided). **c** Western blot analysis for each reprogramming factor in S-/D-OSKM MEFs after Dox treatment for three days. Fold increase in normalized intensity for each reprogramming factor in densitometric analysis is OCT4: 4.61, SOX2: 2.58, KLF4: 3.04, and c-MYC: 3.36. **d** Quantification of NANOG-positive iPSC colonies in S-/D-OSKM MEFs. Data are presented as means ± SD of three independent experiments (two-way ANOVA and Sidak's multiple-comparison test, two-sided). **e** Principal component analysis (PCA) plot of gene expression profiles of mCherry-positive cells at the indicated time points in S-/D-OSKM MEFs, as determined by RNA-seq (S-/D-OSKM MEFs Day 0, $n = 1$; mCherry + MEFs Day 3, $n = 3$; Day 7, $n = 3$; Day 14, $n = 3$). **f** Schematic illustration of the genomic construct of D-OSKM ESCs for tissue-specific expression of OSKM in the kidney or pancreas (upper panel). The lower panel shows the protocol for Dox treatment of chimeric mice. **g** Representative macroscopic images of S-/D-OSKM kidney tumors. Scale bars: 5 mm. **h** Representative histological images and immunostaining for OCT4 and NANOG in S-/D-OSKM pancreatic tumors. Scale bars: 500 μm. **i** Quantification of the OCT4-positive cell area in S-/D-OSKM kidney tumors (normal kidney, $n = 5$; S-OSKM kidney tumor, $n = 18$; D-OSKM kidney tumor, $n = 20$). Data are presented as means ± SD of biologically independent samples (Kruskal–Wallis test and Dunn's multiple-comparison test, two-sided). **j** Representative histological images and OCT4 immunostaining of splenic invasion of D-OSKM pancreatic tumor cells. Scale bars: 1000 μm. **k** qRT-PCR analyses for expression of pluripotency-related genes in S-/D-OSKM kidney tumors. Data are presented as means ± SD of biological triplicates. Relative expression levels to ESCs are shown (one-way ANOVA and Dunnett's multiple-comparison test, two-sided). **l** Representative histological images of S-/D-OSKM tumors in the kidney and pancreas. Scale bars: 500 μm (upper) and 100 μm (lower). **m** Heatmap showing relative expression of uniquely upregulated genes in human testicular germ cell tumors (TGCTs) ($n = 158$). For identification of the uniquely upregulated genes, RNA-seq data from 23 types of human cancer were obtained from the Cancer Genome Atlas (TCGA) in the NIH GDC Data portal (https://portal.gdc.cancer.gov/) (see Fig. 7a). Uniquely upregulated genes in TGCTs were defined as genes that are expressed higher (>2 folds) in TCGTs than any other cancer type. Color range is shown using a $\log_2$ scale. **n** Quantification of the OCT4-positive cell area in D-OSKM kidney tumors after Dox treatment at various doses (2.0 mg/mL, $n = 20$; 0.4 mg/mL, $n = 4$; 0.25 mg/mL, $n = 7$; 0.13 mg/mL, $n = 4$). Data are presented as means ± SD of biologically independent samples (Mann–Whitney test, two-sided).

cells with immature teratoma components (Supplementary Fig. 2i). Furthermore, D-OSKM tumors often exhibited increased expression of genes that are uniquely overexpressed in human GCTs among diverse human cancer types (Fig. 1m). Collectively, these findings imply that transient in vivo induction of D-OSKM resulted in development of cancers with shared characteristics of human GCTs, especially embryonal carcinomas.

To confirm that the observed findings in D-OSKM mice could indeed be attributed to higher levels of OSKM expression, we decreased the dose of Dox during in vivo D-OSKM reprogramming. Treatment with a lower dose of Dox led to a significant reduction in OCT4-positive cell propagation and caused formation of mature teratomas in both the kidney and pancreas of D-OSKM mice (Fig. 1n and Supplementary Fig. 2j,k), confirming that higher levels of OSKM cause the unique phenotypes in D-OSKM mice.

**Propagation of tumor cells that share PGC characteristics in D-OSKM-induced mice.** Previous studies suggested that human GCTs arise from the germ cell lineage, which includes PGCs, and consequently exhibit expression of PGC-related genes[20,21]. Remarkably, expression of a subset of PGC-related genes was modestly upregulated in D-OSKM tumors (Fig. 2a, b). Indeed, OCT4-positive tumor cells in both the kidney and pancreas often coexpressed DAZL, a late gonadal PGC marker (Fig. 2c, d and Supplementary Fig. 2l). In addition, the increased expression of the PGC-related genes was preferentially detected after longer-term OSKM induction (14 days; Fig. 2a, b).

Epigenetic reprogramming, including genome-wide DNA demethylation, takes place in PGCs[27–29]. In particular, erasure of DNA methylation at imprinting control regions (ICRs) occurs exclusively in PGCs during the mammalian life cycle[30,31]. Notably, D-OSKM pancreatic tumors exhibited a significant reduction in DNA methylation at *H19* and *Nespas-Gnasxl* ICRs (Fig. 2e). In sharp contrast, S-OSKM tumors displayed increased methylation levels at *H19* ICR (Fig. 2e), which is consistent with our previous findings[11]. Strikingly, whole-genome bisulfite sequencing (WGBS) and MethylC-seq analysis revealed genome-wide loss of ICR methylation in D-OSKM tumors (Fig. 2f, g and Supplementary Fig. 2m,n). In contrast to the global ICR demethylation, D-OSKM tumors exhibited no reduction in DNA methylation at other

genomic elements (Fig. 2f and Supplementary Fig. 2m,n). These findings suggest that D-OSKM expression in vivo provokes partial activation of the PGC-related transcriptional network and causes genome-wide reduction in ICR methylation in adult somatic cells.

**Trophoblastic differentiation of D-OSKM tumor cells.** We noticed that most D-OSKM tumors in both the kidney and pancreas exhibited extensive hemorrhage; by contrast, hemorrhaging was very rare in S-OSKM tumors (Fig. 1g and Supplementary Fig. 2a). Histological analysis revealed that the hemorrhagic regions in D-OSKM tumors contained clusters of PL-1-positive trophoblast giant cells (TGCs), reminiscent of the histology of the mouse placenta (Fig. 3a–d and Supplementary Fig. 3a,b). PL-1-positive trophoblast giant cells appeared in close proximity to cells expressing CDX2, a marker for precursor cells of trophoblast giant cells (Fig. 3d and Supplementary Fig. 3b). Moreover, CDX2-positive cells never expressed OCT4, recapitulating a pattern observed in preimplantation embryogenesis, when CDX2 and OCT4 are reciprocally expressed during the first cell-fate specification (Fig. 3d)[32]. In agreement with trophoblast differentiation, several trophoblast-related genes were significantly upregulated in D-OSKM kidney tumors (Fig. 3e). As with PGC-related gene expression, trophoblast-related gene expression became apparent following longer-term induction of OSKM (14 days; Fig. 3e), suggesting that trophoblast differentiation coincided with emergence of PGC-associated features. Considering that human embryonal carcinomas occasionally contain extraembryonic cells and exhibit features of totipotency, the presence of trophoblast giant cells in D-OSKM tumors provides additional evidence that D-OSKM tumors share properties with human embryonal carcinomas.

To determine whether the trophoblasts in D-OSKM tumors were derived from OCT4/NANOG-positive tumor cells, we next performed a lineage-tracing experiment for *Nanog*-expressing cells (Fig. 3f). Because of a technical limitation, we utilized S-OSKM alleles to induce kidney tumors; the S-OSKM tumors also contained trophoblast giant cells, albeit at a lower frequency (Fig. 3b). Notably, we observed reporter-positive trophoblast giant cells in OSKM-induced kidney tumors (Fig. 3g), implying

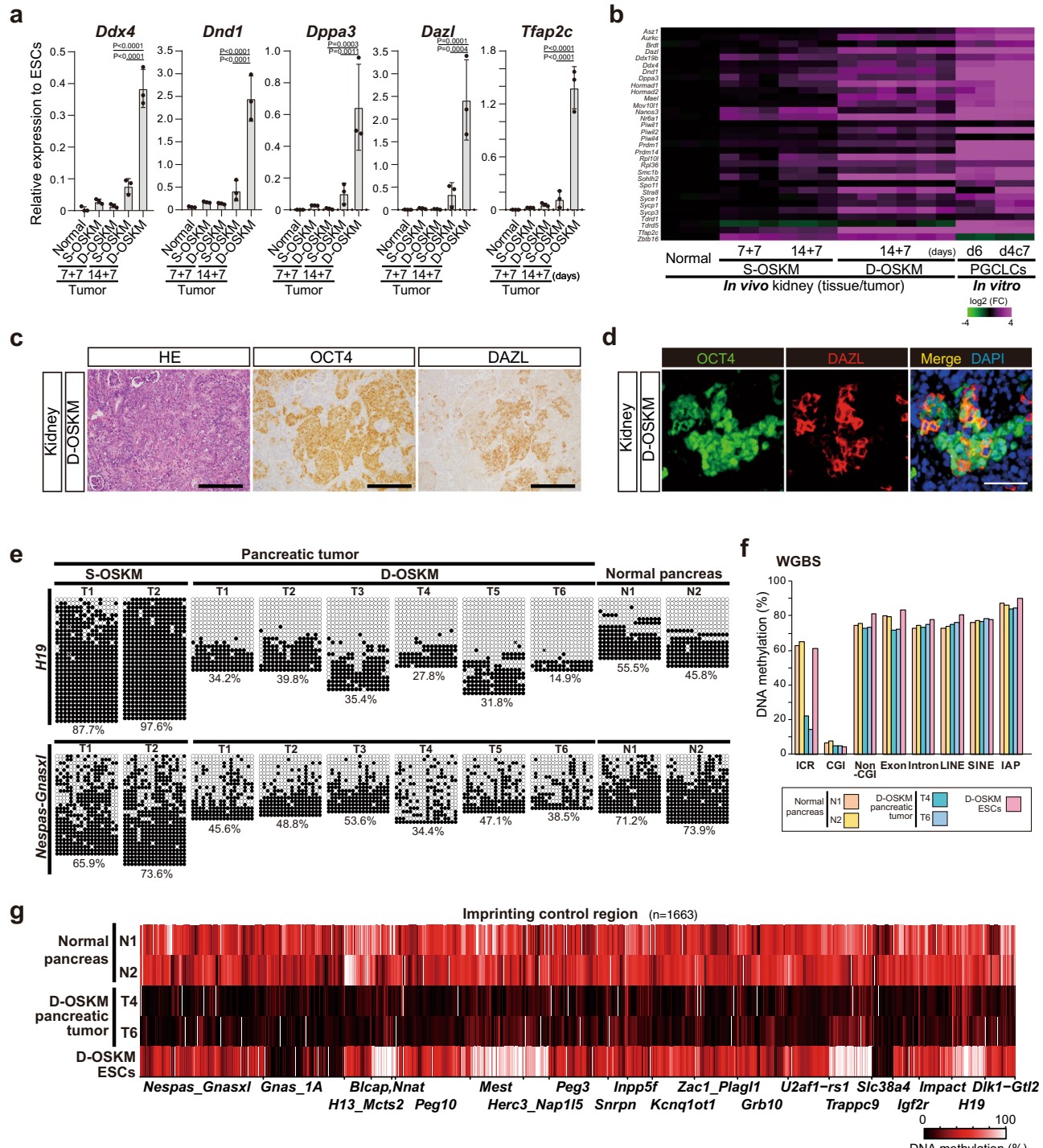

**Fig. 2 PGC-associated signatures in D-OSKM-induced tumors. a** qRT-PCR analyses of PGC-related gene expression in S-/D-OSKM kidney tumors. Data are presented as means ± SD of biological triplicates. Relative expression levels to ESCs are shown (one-way ANOVA and Dunnett's multiple-comparison test, two-sided). **b** Heatmap showing relative expression of germ line-related genes[73, 74] in S-/D-OSKM kidney tumors. PGC-like cells (PGCLCs) were derived from ESCs as described previously[39, 74]. Color range is shown using a log$_2$ scale. **c** Representative histological images and immunostaining for OCT4 and DAZL of D-OSKM kidney tumors. Scale bars: 200 μm. **d** Immunofluorescence staining for OCT4 and DAZL in D-OSKM kidney tumors. Scale bars: 50 μm. **e** DNA methylation analysis of individual CpG sites at the *H19* ICR and *Nespas-Gnasxl* ICR in D-OSKM pancreatic tumors. **f** DNA methylation percentage at various genetic elements in D-OSKM pancreatic tumors, measured by whole-genome bisulfite sequencing (WGBS). CGI, CpG island; non-CGI, non-CpG island. **g** Heatmap showing global ICR methylation in D-OSKM pancreatic tumors, as determined by MethylC-seq analysis. *n* indicates the number of examined CpG sites.

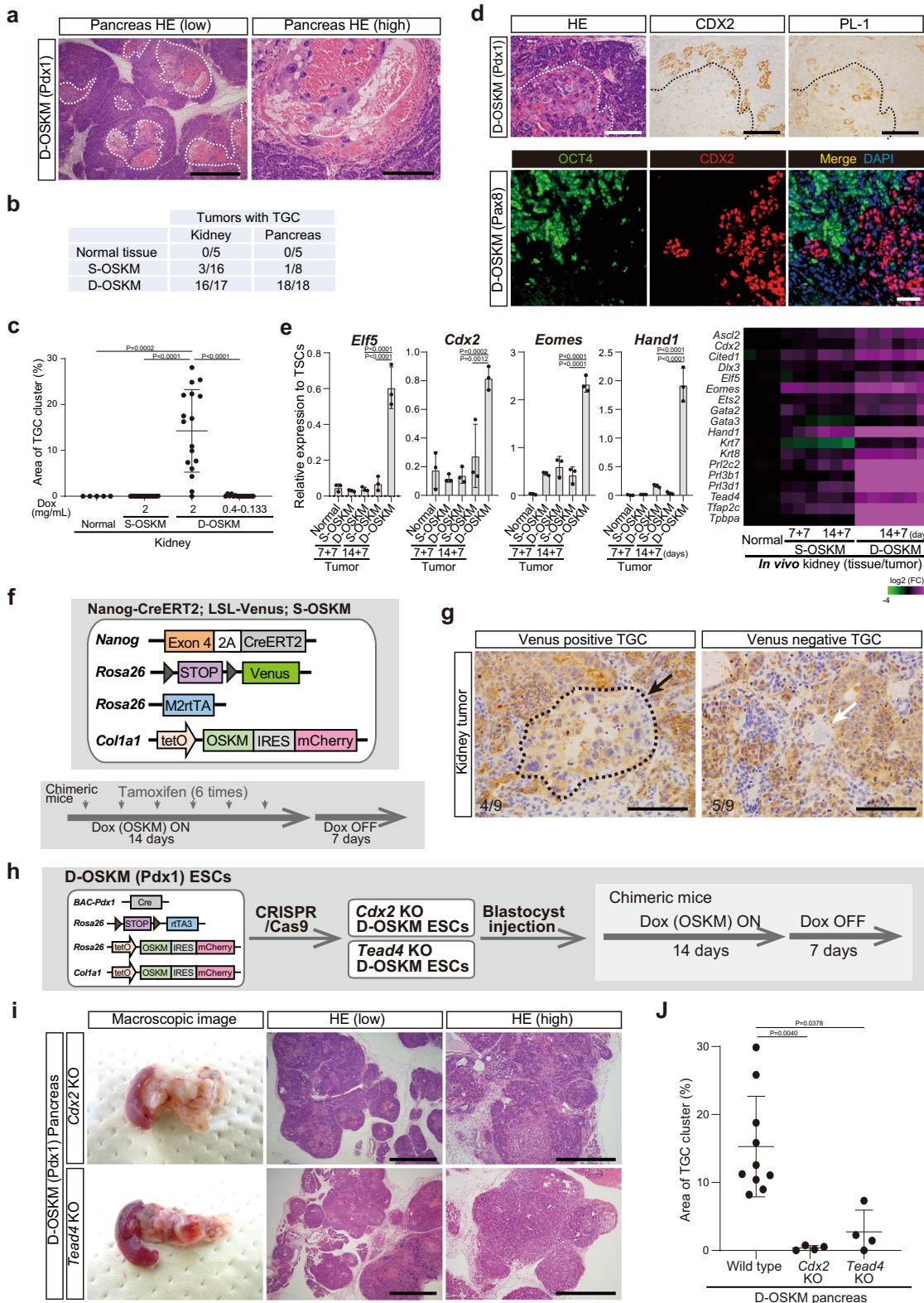

that *Nanog*-expressing cells give rise to trophoblast giant cells during in vivo reprogramming.

In preimplantation embryogenesis, the first cell-fate specification into an either embryonic or extraembryonic lineage is mediated through the *Tead4–Cdx2* axis in totipotent blastomeres and 8-cell embryos. Indeed, genetic ablation of either *Tead4* or *Cdx2* causes developmental defects in trophoblast differentiation in preimplantation embryos[33,34]. To investigate the underlying mechanisms of

trophoblastic differentiation in D-OSKM tumors, we next established D-OSKM chimeric mice deficient for *Tead4* or *Cdx2* (Fig. 3h and Supplementary Fig. 3c,d). Although *Tead4*- or *Cdx2*-knockout (KO) D-OSKM mice developed tumors, the number of trophoblast giant cells was remarkably reduced (Fig. 3i, j and Supplementary Fig. 3d,e), suggesting that the machinery involved in trophoblastic differentiation in D-OSKM tumors is similar to that in preimplantation embryogenesis.

**Fig. 3 Trophoblastic differentiation of D-OSKM tumor cells. a** Representative histological images of D-OSKM pancreatic tumors containing trophoblast giant cell (TGC) clusters. Dotted lines indicate TGC clusters (>3 cells/high-power field [×400]). Scale bars: 1000 μm (left), 200 μm (right). **b** Incidence of tumors containing trophoblast giant cells in S-/D-OSKM chimeric mice. **c** Quantification of TGC cluster area in S-/D-OSKM kidney tumors (normal kidney, $n = 5$; S-OSKM kidney tumor, $n = 16$; D-OSKM kidney tumor following Dox treatment at various doses [2.0 mg/mL, $n = 17$; 0.4 mg/mL, $n = 4$; 0.25 mg/mL, $n = 7$; 0.13 mg/mL, $n = 4$]). Data are presented as means ± SD of biologically independent samples (Kruskal–Wallis test and Dunn's multiple-comparison test, two-sided). **d** Representative histological images and immunostaining for CDX2 and PL-1 of D-OSKM pancreatic tumors (upper panels). Dotted lines indicate TGC clusters. Scale bars: 200 μm. Immunofluorescence staining for OCT4 and CDX2 of D-OSKM kidney tumors (lower panels). Scale bars: 50 μm. **e** (Left) qRT-PCR analyses of trophoblast-related gene expression in S-/D-OSKM kidney tumors. Data are presented as means ± SD of biological triplicates. Relative expression levels to TSCs are shown (one-way ANOVA and Dunnett's multiple-comparison test, two-sided) (Right) Heatmap showing relative expression of trophoblast-related genes in S-/D-OSKM kidney tumors. Color range is shown using a $\log_2$ scale. **f** Lineage-tracing system for *Nanog*-expressing cells. Mice were treated with tamoxifen (2.0 mg) six times every other day during Dox treatment. **g** Immunostaining for Venus in kidney tumors ($n = 4$). Arrows indicate TGCs. Dotted lines indicate the TGC cluster. Scale bars: 200 μm. **h** Schematic illustration of an experimental protocol for in vivo D-OSKM reprogramming of somatic cells deficient for *Cdx2* or *Tead4*. **i**, Representative macroscopic and histological images of *Cdx2* KO and *Tead4* KO D-OSKM pancreatic tumors. Scale bars: 1000 μm (middle), 500 μm (right). **j** Quantification of TGC cluster area in control ($n = 10$), *Cdx2* KO ($n = 4$), and *Tead4* KO ($n = 4$) D-OSKM pancreatic tumors. Data are presented as means ± SD of biologically independent samples (Kruskal–Wallis test and Dunn's multiple-comparison test, two-sided).

Finally, we investigated whether the expression of PGC-related genes and trophoblast differentiation was specific to in vivo reprogramming. To this end, we compared expression of pluripotency-, PGC-, and trophoblast-related genes between in vitro (MEF) and in vivo (kidney) D-OSKM reprogramming. We found that the increased expression of PGC- and trophoblast-related genes was predominantly detectable during the in vivo reprogramming process (Supplementary Fig. 3f).

**Derivation of D-OSKM tumor-derived PSCs with the potential to differentiate toward the extraembryonic lineage**. Above, we showed that in vivo D-OSKM reprogramming results in propagation of OCT4-positive cells that also expressed pluripotency-related genes. Therefore, we next tried to establish PSCs from D-OSKM tumor cells (Fig. 4a). To this end, we minced D-OSKM kidney tumors and cultured them in ESC medium. A number of dome-shaped PSC-like colonies emerged at 4–6 days after seeding of D-OSKM kidney tumor cells (Supplementary Fig. 4a). We then subcloned these cells to establish three independent iPSC-like cell lines from each kidney tumor (Supplementary Fig. 4a). D-OSKM tumor-derived iPSC-like cells expressed pluripotency-related genes at levels similar to those in PSCs (Supplementary Fig. 4b). As shown in our previous study[11], iPSCs capable of forming mature teratomas were derived from S-OSKM tumor cells (S-iPSCs; Supplementary Fig. 4c). Similarly, D-OSKM tumor-derived iPSC-like cells gave rise to tumors containing a mature teratoma component (Supplementary Fig. 4c), demonstrating that D-OSKM iPSC-like cells were also PSCs (D-iPSCs). Notably, however, the majority of D-iPSC tumors exhibited hemorrhage, which phenocopied D-OSKM tumors (Fig. 4b). Consistent with this, clusters of PL-1-positive trophoblast giant cells were observed at high frequency in D-iPSC tumors, while they are hardly detected in S-iPSC tumors (Fig. 4b, c and Supplementary Fig. 4d). However, the number of trophoblast giant cell clusters in D-iPSC tumors decreased after prolonged passage of D-iPSCs in vitro (Supplementary Fig. 4d,e), indicating that ESC culture conditions could not maintain the expanded differentiation potential of D-iPSCs for longer periods of time.

In addition, we performed embryoid body (EB) formation assays. Upon EB induction, D-iPSC EBs formed buds at the periphery that contained CDX2-positive cells and exhibited upregulation of trophoblast-related genes (Fig. 4d, e and Supplementary Fig. 4f). Consistent with this, trophoblast giant cell-like cells appeared within 4 days after attachment of D-iPSC EBs to gelatin-coated dishes (Supplementary Fig. 4g). In sharp contrast to trophoblastic differentiation, PGC-related

transcriptional signatures were not detected in D-iPSC EBs (Supplementary Fig. 4h), which is in agreement with the fact that embryonic germ cells (EG cells: PSCs derived from PGCs) exhibit transcriptional signatures of ESCs rather than PGCs[35]. To further examine the extended differentiation capacity of D-iPSCs at the single-cell level, we isolated single D-iPSCs by FACS and established 24 subclones (Supplementary Fig. 4i). Notably, the majority of teratomas from D-iPSC subclones contained a cluster of PL-1-positive trophoblast giant cells (Supplementary Fig. 4i), demonstrating that a single D-iPSC eventually differentiates into both embryonic and extraembryonic lineage cells.

Based on the above findings, we next tested the in vivo differentiation capacity of D-iPSCs in preimplantation embryos. We injected eight to ten GFP-labeled D-iPSCs into eight-cell embryos, and then examined the chimeric contribution in blastocysts after 24–36 h of in vitro culture. As expected, all cell types (ESCs, S-iPSCs, and D-iPSCs) contributed to the ICM of chimeric blastocysts (Fig. 4f). Notably, D-iPSCs also contributed to the mural trophectoderm (TE) layers (Fig. 4f and Supplementary Fig. 4j). To further assess the differentiation potential of D-iPSCs, we transferred chimeric blastocysts into the uterus and examined the chimeric concepti at E13.5. D-iPSC-derived cells were present in multiple organs of E13.5 embryos (10 out of 16 embryos; Fig. 4g and Supplementary Fig. 4k). Furthermore, the chimeric mice were born and grew normally without an obvious phenotype, until at least nine months of age ($n = 3$; Fig. 4h). We also observed GFP-positive cells in the E13.5 placenta, albeit at a low frequency (two out of 16 placentae; Fig. 4g). Immunohistochemical analyses confirmed the contribution of D-iPSC-derived cells to the spongiotrophoblast layers (Supplementary Fig. 4l). Taken together, these findings show that D-iPSCs acquire bidirectional differentiation capacity, i.e., the ability to differentiate into both embryonic and extra-embryonic lineages.

Considering that erasure of ICR methylation occurs exclusively at late PGCs, and that the erased methylation cannot be recovered in PSC-derived somatic cells[31], we can trace cells that have undergone ICR demethylation. Therefore, we next analyzed DNA methylation status at ICRs in D-iPSCs derived from pancreatic tumors. Notably, in contrast to increased methylation levels at *H19* ICR in S-iPSCs, the majority of D-iPSCs exhibited a significant reduction in DNA methylation levels at both *H19* and *Nespas-Gnasxl* ICRs (Fig. 4i). Consistent with this, expression of *H19* was increased in D-iPSCs (Supplementary Fig. 5a). On the basis of these findings, we conclude that D-iPSCs are indeed derived from tumor cells that harbor unique characteristics of PGCs.

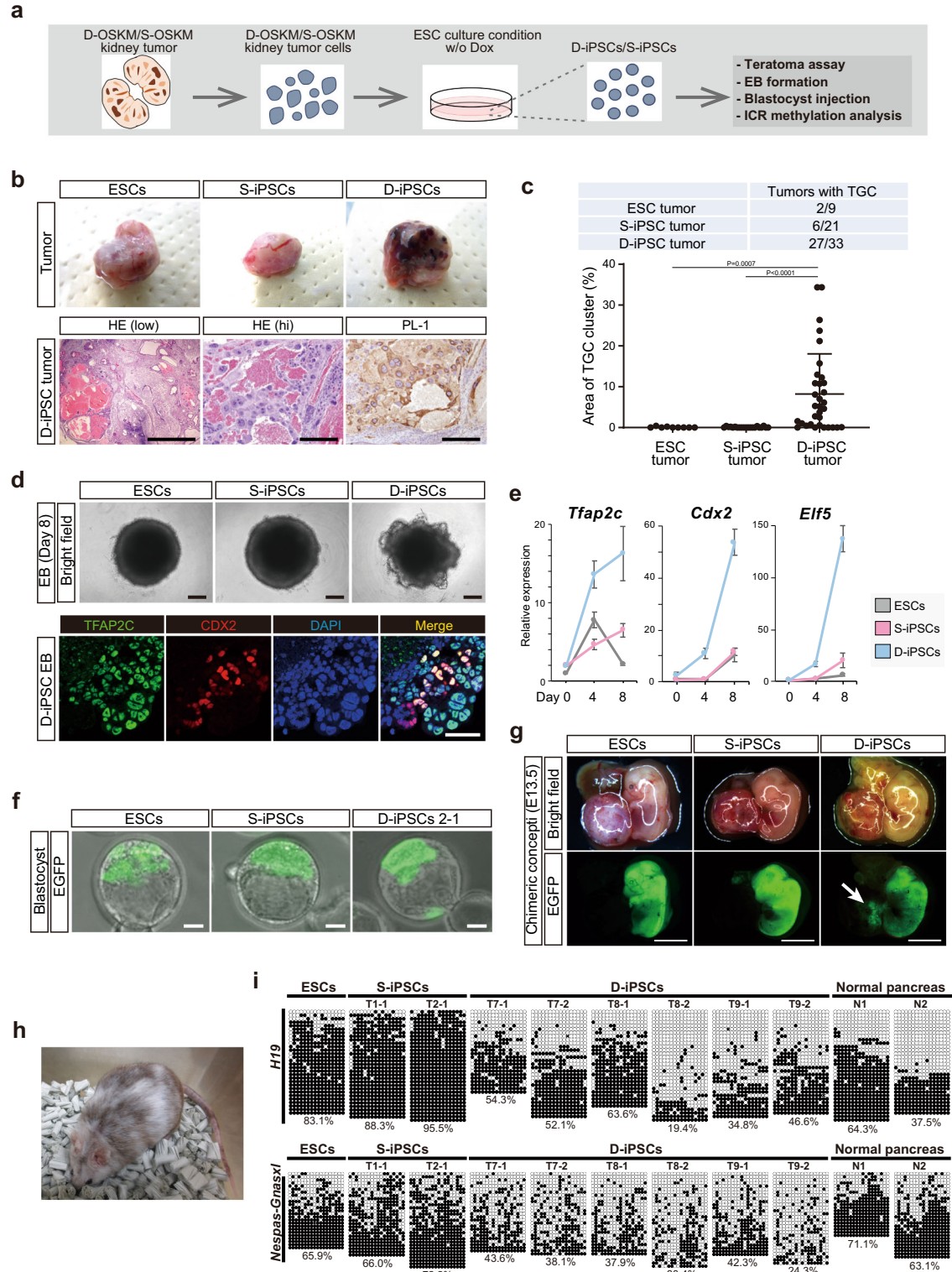

2C-like cells (2 CLCs), a rare population of ESCs, have extended differentiation potential, i.e., they can differentiate into both embryonic and extraembryonic lineage cells[36]. In mechanistic terms, DUX is responsible for the unique transcriptional signatures in the 2C-like state[37,38]. Other studies succeeded in establishing PSCs with expanded differentiation potential from ESCs or blastomeres by chemical inhibition of multiple signaling pathways[22,23]. Notably, D-iPSCs did not exhibited higher levels of DUX expression, nor did their global transcriptional profiles share

features with these cell populations (Supplementary Fig. 5b,c). Additionally, the expression profiles of D-iPSCs did not exhibited a significant correlation with totipotent cells but showed the highest correlation with ICM-relative preimplantation embryos at various developmental stages (Supplementary Fig. 5d).

**Increased DMRT1 chromatin accessibility is responsible for expanded differentiation potential in PSCs.** Given that the bidirectional differentiation capacity of D-iPSCs gradually

**Fig. 4 Derivation of D-OSKM tumor-derived PSCs with the potential to differentiate toward the extraembryonic lineage. a** Schematic illustration of iPSC derivation from S-/D-OSKM kidney tumor cells (S-/D-iPSCs). A total of nine S-/D-iPSC clones were established from three independent kidney tumors. **b** Representative macroscopic (upper) and histological (lower) images of ESC-, S-iPSC-, and D-iPSC-derived subcutaneous tumors in immunocompromised mice (upper panels). Immunostaining for PL-1 is shown in the lower panel. Scale bars: 500 μm (left), 200 μm (middle and right). **c** Incidence of tumors containing TGCs (upper) and quantification of TGC cluster area in ESC ($n = 9$), S-iPSC ($n = 21$), and D-iPSC tumors ($n = 33$). Nine independent S-/D-iPSC clones were used. Data are presented as means ± SD (Kruskal–Wallis test and Dunn's multiple-comparison test, two-sided). **d** Representative images of ESC-, S-iPSC-, and D-iPSC-derived EBs (upper panels). Lower panels show immunofluorescence staining for TFAP2C and CDX2 of D-iPSC EB at differentiation Day 8. Scale bars: 200 μm (upper), 50 μm (lower). **e** qRT-PCR analyses for expression of trophoblast-related genes during EB formation. Data are presented as means ± SD of biological triplicates. The mean expression level at differentiation Day 0 ESCs was defined as 1. **f** Representative images of blastocysts injected with GFP-labeled ESCs, S-iPSCs, or D-iPSCs. Scale bars: 20 μm. **g** Representative images of E13.5 chimeric concepti. The chimeric blastocysts were transferred into the uterus of pseudopregnant ICR mice. Distinct GFP signals were observed in two of 16 placentae of D-iPSC chimeric concepti (arrow). Scale bars: 5 mm. **h** Adult D-iPSC chimeric mouse at 10 months of age. **i** DNA methylation analysis of individual CpG sites at the *H19* and *Nespas-Gnasxl* ICRs in D-iPSCs.

disappeared after prolonged culture (Supplementary Fig. 4d,e), we speculated that epigenetic regulation, rather than genetic alterations, contributes to the unique properties of D-iPSCs. To understand the epigenetic landscape associated with the expanded differentiation potential of D-iPSCs, we compared chromatin accessibility using ATAC-seq (assay for transposase-accessible chromatin sequencing). To this end, we identified 270 peaks that were uniquely accessible in early-passage D-iPSCs, the only cells tested that have bidirectional differentiation potential (Supplementary Fig. 6a). Notably, motif analysis revealed that DMRT1/6 binding motifs were significantly enriched in the D-iPSC-specific peaks (Fig. 5a, b and Supplementary Fig. 6b). ATAC-qPCR analysis confirmed the increased accessibility at representative loci containing the DMRT1/6-binding motif in multiple D-iPSC clones ($n = 2$, Fig. 5b and Supplementary Fig. 6c). DMRT1 is a transcription factor distinctively expressed in the gonadal cells, including PGCs, which is in agreement with our findings that D-OSKM tumor cells exhibit PGC-related features. Indeed, *Dmrt1*, but not *Dmrt6*, was modestly upregulated in both D-iPSCs and D-OSKM tumors (Fig. 5c, d and Supplementary Fig. 6d). Most notably, DMRT1 expression was exclusively localized in DAZL-positive cells that also express OCT4 in D-OSKM tumors (Fig. 5e), indicating that expression of DMRT1 coincides with DAZL expression, which is reminiscent of expression patterns in late PGCs[39].

To assess the impact of elevated DMRT1 expression in OCT4-expressing cells, we next established ESCs carrying a Dox-inducible *Dmrt1* (Fig. 5f and Supplementary Fig. 7a). Upon EB induction, *Dmrt1* expression caused a rapid reduction in pluripotency-related gene expression (Supplementary Fig. 7b), in accordance with previous studies demonstrating that *Dmrt1* negatively regulates pluripotency-related genes[40,41]. Moreover, *Dmrt1* expression led to the formation of buds containing CDX2-positive cells at the periphery of EBs (Fig. 5g, h), as observed in D-iPSC EBs. We also found that *Dmrt1* expression converted ESC colonies to an epithelial cell-like morphology only in the presence of FGF4 or FGF2 in 2D cultures (Supplementary Fig. 7c), indicative of FGF-dependent differentiation; this is also consistent with previous findings that FGF-MAPK signaling is important for trophoblast differentiation and proliferation[42,43]. However, the majority of epithelial-like cells did not yet express CDX2 at Day 4 of *Dmrt1* induction (Supplementary Fig. 7d,e). These results suggest that DMRT1 expression induces CDX2 through FGF-dependent but indirect mechanisms. Together with the enrichment of the DMRT1-binding motif at the unique chromatin landscape of D-iPSCs, these results raised the further possibility that DMRT1-mediated epigenetic reprogramming confers trophoblastic competence. To test this hypothesis, we transiently induced *Dmrt1* in ESCs and examined the differentiation propensity of the *Dmrt1*-induced cells (Supplementary Fig. 7f,g). Because FGF

secretion from mesenchymal cells plays a crucial role in stem cell homeostasis in the testis[44], we inoculated the *Dmrt1*-induced cells into the testis. There, *Dmrt1*-induced cells gave rise to teratomas containing clusters of trophoblast giant cells (Fig. 5i), providing additional evidence that DMRT1-mediated reprogramming is involved in eliciting trophoblast differentiation capacity.

**DMRT1 drives in vivo reprogramming and propagation of GCT-like tumor cells.** Finally, we established D-OSKM chimeric mice deficient for *Dmrt1* and examined the role of DMRT1 during in vivo D-OSKM reprogramming (Fig. 6a). Previous studies demonstrated that knockdown of *Dmrt1* does not have a significant effect on the efficiency of in vitro reprogramming[45,46]. Additionally, genetic ablation of *Dmrt1* did not cause significant differences in ESC proliferation in vitro or tumorigenicity in vivo (Supplementary Fig. 8a–c). Strikingly, however, *Dmrt1* KO significantly decreased the tumor area in D-OSKM chimeric mice (Fig. 6b, c and Supplementary Fig. 8d). Consistent with this, *Dmrt1* KO D-OSKM tumors had fewer OCT4- and Ki67-positive cells (Fig. 6b, d, e and Supplementary Fig. 8d). Moreover, DAZL-positive cells were nearly absent in *Dmrt1* KO D-OSKM tumors (Fig. 6f and Supplementary Fig. 8d). Furthermore, we failed to establish PSCs from *Dmrt1* KO D-OSKM tumors (Fig. 6g). We also found that *Dmrt1* KO led to a reduced number of CDX2-positive cells and downregulation of trophoblast-related genes in D-OSKM tumors (Fig. 6h, i and Supplementary Fig. 8d,e). Collectively, these findings indicate that DMRT1-mediated in vivo reprogramming drives propagation of D-OSKM tumor cells, which can give rise to PSCs with expanded differentiation capacity into trophoblasts (Fig. 6j).

**DMRT1-mediated reprogramming provides a potential therapeutic target for human germ cell tumors.** To assess the possible involvement of DMRT1-mediated in vivo reprogramming in human GCT development, we first examined expression patterns of DMRT1. We found that DMRT1 is highly expressed in germ cell neoplasia in situ (GCNIS), a premalignant lesion of various types of testicular germ cell tumors (TGCTs), while it is not expressed in most cancer types (Supplementary Fig. 9a,b). We next examined the transcriptional correlation between human TGCTs and human germ-line cells at various developmental stages. Notably, TGCTs, particularly nonseminomas, were more strongly correlated with ESCs than PGCs (Fig. 7a), which support the assumption that reprogramming toward PSCs is involved during human GCT development. Notably, ATAC-seq identified the DMRT1/6 motif as one of the most enriched motifs in TGCTs (Fig. 7b and Supplementary Fig. 9c,d). Considering that DMRT1 is expressed in germ cells, as well as the premalignant lesions, we

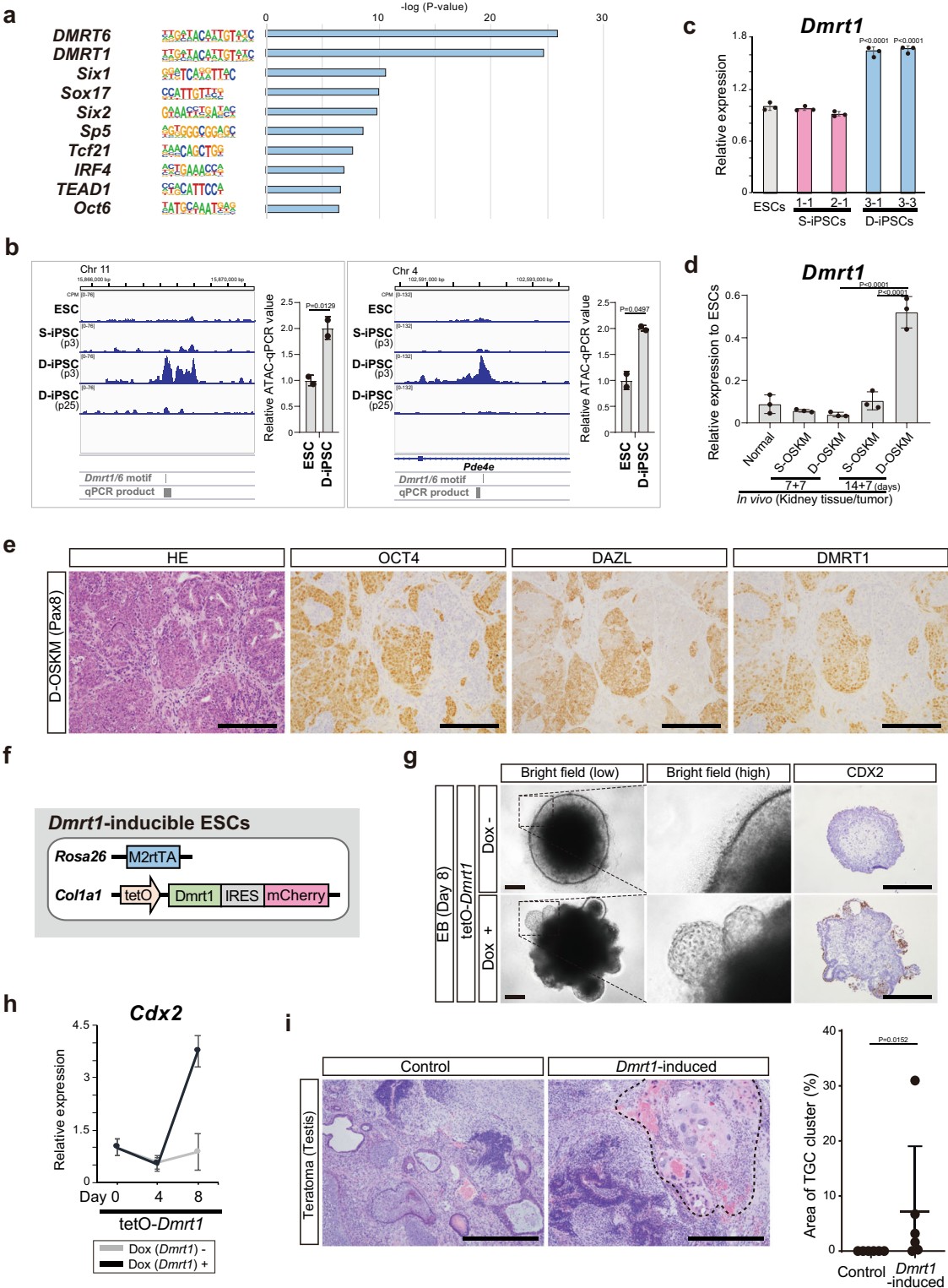

propose that GCT cells may retain the memory of the DMRT1-mediated epigenetic landscape.

In contrast to a crucial role of DMRT1-mediated reprogramming on GCT-like tumor development, DMRT1 induction promoted trophoblast differentiation of OCT4-expressing PSCs. Therefore, we speculated that this unique chromatin accessibility might be a vulnerability of GCT cells. As expected, induction of *Dmrt1* resulted in a growth-arrest phenotype in all GCT cell lines

tested (Fig. 7c and Supplementary Fig. 10a,b). Remarkably, *Dmrt1* expression led to trophoblastic differentiation in NCCIT, an embryonal carcinoma cell line (Fig. 7d, e). By contrast, induction of *Dmrt1* had little effect on cancer cell growth of other cancer types (Supplementary Fig. 10c). Since elevated chromatin accessibility at DMRT1-binding sites is a unique epigenetic feature of GCTs, it may serve as a possible therapeutic target for GCTs.

**Fig. 5 Elevated DMRT1 chromatin accessibility is associated with the extended differentiation potential of PSCs. a** Motif enrichment analysis of D-iPSC-specific enriched peaks in ATAC-seq ($n = 270$), performed using HOMER (Fisher's exact test, two-sided). **b** Representative specific peaks in D-iPSCs and ATAC-qPCR analysis. Genomic regions containing DMRT1/6-binding motifs[75, 76] and qPCR products are shown. Data are presented as means ± SD of biological duplicates. The mean ATAC-qPCR value of ESCs was defined as 1 (*t*-test, two-sided). **c** qRT-PCR analysis of *Dmrt1* expression in ESCs, S-iPSCs, and D-iPSCs. Data are presented as means ± SD of technical triplicates. The mean expression level of ESCs was defined as 1 (one-way ANOVA, two-sided). **d** qRT-PCR analysis for expression of *Dmrt1* in S-/D-OSKM kidney tumors. Data are presented as means ± SD of biological triplicates. Relative expression levels to ESCs are shown (one-way ANOVA and Dunnett's multiple-comparison test, two-sided). **e** Representative histological images and immunostaining for OCT4, DAZL, and DMRT1 of D-OSKM kidney tumors. Scale bars: 200 μm. **f** Schematic illustration of the genomic construct of *Dmrt1*-inducible ESCs. **g** Representative images and immunostaining for CDX2 in *Dmrt1*-inducible EBs on differentiation Day 8. Scale bars: 200 μm (left), 500 μm (right). **h** qRT-PCR analysis of *Cdx2* expression during EB differentiation. Data are presented as means ± SD of biological triplicates. The mean expression level in ESCs on differentiation Day 0 was defined as 1. **i** Representative histological images of tumors developed after inoculation of *Dmrt1*-induced cells into the testis of 6-week-old C57BL/6 N mice. Dotted lines indicate the TGC cluster. Right panel represents quantification of TGC cluster area in tumors (control, $n = 6$; *Dmrt1*-induced cells, $n = 6$). Data are presented as means ± SD of biologically independent samples. Scale bars: 500 μm (Mann–Whitney test, two-sided).

## Discussion

OSKM expression in vivo has a variety of consequences, including cancer development, cellular senescence, tissue regeneration, and organismal rejuvenation[10–13]. Here, we showed that in vivo expression of D-OSKM causes increased expression of PGC-related genes and provokes the genome-wide ICR demethylation in adult somatic cells. These phenotypes were observed in pancreas and kidney, indicating that the cellular context of the starting cells is not critical for the induction of PGC-associated signatures. However, the increased expression of PGC-related genes was not evident after seven days of in vivo D-OSKM induction, nor during in vitro D-OSKM reprogramming, indicating that longer periods of OSKM expression, along with features of the in vivo environment, are needed for the expansion of cells with PGC-associated signatures. Together with the fact that the cell-fate destination of OSKM-induced cells is affected by extracellular environments[8,9], this suggests that the in vivo environment, comprising multiple cell types, facilitates coordinated but distinct cellular responses, depending on the levels and periods of OSKM expression, eventually leading to various outcomes, including the acquisition and propagation of cells with totipotency-like features. Precise understanding of in vivo reprogramming at the single-cell level might uncover the fundamental basis of interplay among multiple cell types, which might further contribute to a better understanding of various biological functions at the organismal level.

GCTs are neoplasms arising from the germ cell-lineage cells in the gonads and extragonadal sites[20,21]. A current consensus holds that embryonic germ-line cells, especially mismigrated PGCs or differentiation/maturation-arrested PGCs, are the cells of origin for GCTs. Based on phenotypic observations that several subtypes of GCTs occasionally exhibit differentiation into extraembryonic lineage cells, it has also been proposed that reprogramming into the early preimplantation totipotent cell-like state occurs during GCT development[21]. Here, we showed that the transcriptional profiles of human GCTs, particularly nonseminomas, were more strongly correlated with those of ESCs than with those of embryonic germ cell progenitors. Moreover, transient induction of D-OSKM in vivo caused development of GCT-like cancers with the potential to differentiate into trophoblasts. Although the genetic lineage tracing with a PGC-specific reporter is required to unequivocally demonstrate that the PGC-related reprogramming takes place before trophoblast differentiation at a cellular level, these results support the notion that reprogramming indeed drives development of a subgroup of GCTs. These results are also consistent with the assumption that a gain of isochromosome 12p, which includes *NANOG*, promotes the progression of human GCTs[47,48] by stimulating reprogramming[49,50]. Remarkably, D-OSKM tumors exhibit global hypomethylation at ICRs, a distinctive epigenetic feature of human TGCTs[51]. Global ICR hypomethylation in GCTs has been explained based on the epigenetic features of the cells of origin, i.e., late PGCs have lost ICR methylation. Considering the results in this study, it is also conceivable that reprogramming is responsible for the ICR demethylation in human GCTs. We also showed that GCT-like D-OSKM tumor cells gave rise to iPSCs that contributed to nonneoplastic cells in adult mice, suggesting that GCT-like cancer development does not require genetic transformation, but instead depends on epigenetic alterations[11]. This is consistent with the seminal work by Minz demonstrating that teratocarcinoma cells, which were derived by grafting early mouse embryos into the testes and maintained as an ascite tumor, contributed to somatic cells of adult mice after injection into blastocysts[52]. Together, these findings further underscore the central role of epigenetic regulation in the development of GCTs. Collectively, we propose that the in vivo D-OSKM reprogramming system offers a novel mouse model for studying human GCT development.

Given that acquisition of totipotency is a critical step in the creation of new individuals, which in turn provides a driving force for genetic diversity and evolution, how embryonic lineage cells acquire totipotency is a fundamental question in mammalian biology. Despite their bidirectional differentiation potential, D-iPSCs exhibited a global transcriptional profile distinct from those of previously reported PSCs with similar differentiation propensity. Notably in this regard, we found that *Dmrt1* expression was generally higher in cell populations with totipotency-like features (Supplementary Fig. 10d). It is possible that DMRT1-mediated reprogramming is associated with acquisition of expanded differentiation potential in embryonic lineage cells. Intriguingly, we also found that DMRT1-binding sites in E13.5 gonadal cells were most accessible in the morula stage during preimplantation embryogenesis, albeit in the absence of robust *Dmrt1* expression (Supplementary Fig. 10e)[53–55]. Given that the initial trophoblastic differentiation is manifested around the morula stage, the elevated accessibility might play a role in the acquisition of totipotency in preimplantation embryos as well.

In conclusion, in vivo expression of OSKM provokes PGC-associated features in adult somatic cells and promotes the development of human GCT-like cancers that exhibit trophoblast differentiation. DMRT1-mediated reprogramming plays a crucial role in the propagation of GCT-like tumor cells that have the propensity to differentiate into trophoblasts. Our results uncover a novel intermediate state during in vivo reprogramming, which also confers unique differentiation propensity on in vivo PSCs.

## Methods
### BAC and vector construction for gene targeting
*Pax8-Cre, Pdx1-Cre BAC.* A 2.8-kb fragment of *Cre-pA-PGK-Bsd-pA* with 50 bp homology arms was established using KAPA HiFi HotStart ReadyMix (KAPA Biosystems). This fragment was inserted into the start codon of *Pax8* or *Pdx1* in a

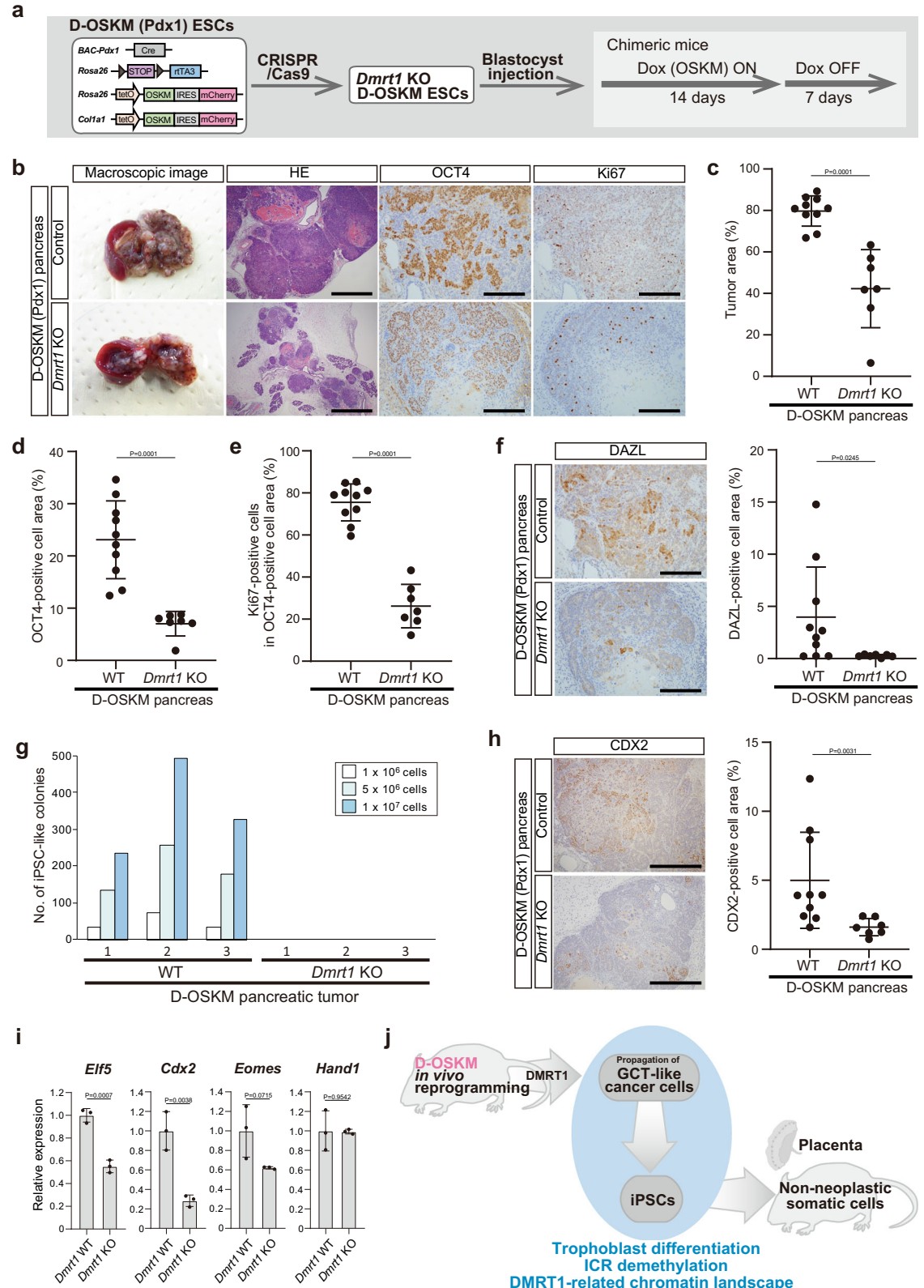

BAC containing the locus (BACPAC Resources Center) using the Red/ET BAC-recombination system (GeneBridges).

*Rosa26::tetO-OSKM-ires-mCherry* targeting vector. A 5.1-kb polycistronic cassette containing *Oct4-P2A-Sox2-T2A-Klf4-E2A-c-Myc*[56] was cloned using KAPA HiFi HotStart ReadyMix (KAPA Biosystems). This cassette was inserted into pCR8-

GW-TOPO using TOPO cloning technology (Invitrogen), transferred into the *Rosa26 tetO-attR1-ccdB-attR2-ires-mCherry* targeting vector using the Gateway technology (Thermo Fisher Scientific), and used as a targeting vector.

*Rosa26 LSL-venus* targeting vector. *Venus* with Kozak sequence (723 bp) was cloned using KOD-Plus-Neo (TOYOBO). This fragment was inserted into pCR8-

**Fig. 6 DMRT1 drives in vivo reprogramming and propagation of GCT-like tumor cells. a** Schematic illustration of the experimental protocol for in vivo D-OSKM reprogramming of somatic cells deficient for *Dmrt1*. **b** Representative macroscopic and histological images of pancreatic tumors in *Dmrt1*-KO D-OSKM chimeric mice. Immunostaining for OCT4 and Ki67 in pancreatic tumors is shown in the right panels. Scale bars: 1000 μm (left), 200 μm (middle and right). **c–e** Quantification of tumor area (c), OCT4-positive cell area in the tumor (d), and Ki67-positive cell ratio in the OCT4-positive cell area (e) in the pancreas. D-OSKM mice that developed pancreatic tumors were evaluated in this study (control, n = 10; *Dmrt1* KO18, n = 2; *Dmrt1* KO21, n = 5). Data are presented as means ± SD of biologically independent samples (Mann–Whitney test, two-sided). **f** Immunostaining for DAZL and quantification of the DAZL-positive cell area in pancreatic tumors (control, n = 10; *Dmrt1* KO18, n = 2; *Dmrt1* KO21, n = 5). Data are presented as means ± SD of biologically independent samples (Mann–Whitney test, two-sided) scale bars: 200 μm. **g** Derivation efficiency of iPSC-like colonies from pancreatic tumors (control, n = 3; *Dmrt1* KO21, n = 3). Colony counting was performed at Day 7. **h** Immunostaining for CDX2 and quantification of the CDX2-positive cell area in pancreatic tumors (control, n = 10; *Dmrt1* KO18, n = 2; *Dmrt1* KO21, n = 5). Data are presented as means ± SD of biologically independent samples (Mann–Whitney test, two-sided) scale bars: 500 μm. **i** qRT-PCR analyses of trophoblast-related gene expression in pancreatic tumors. Data are presented as means ± SD of biological triplicates. The mean expression level of control D-OSKM pancreatic tumors was defined as 1 (*t*-test, two-sided). **j** Schematic representation of in vivo D-OSKM reprogramming and derivation of PSCs with the expanded differentiation potential into trophoblasts.

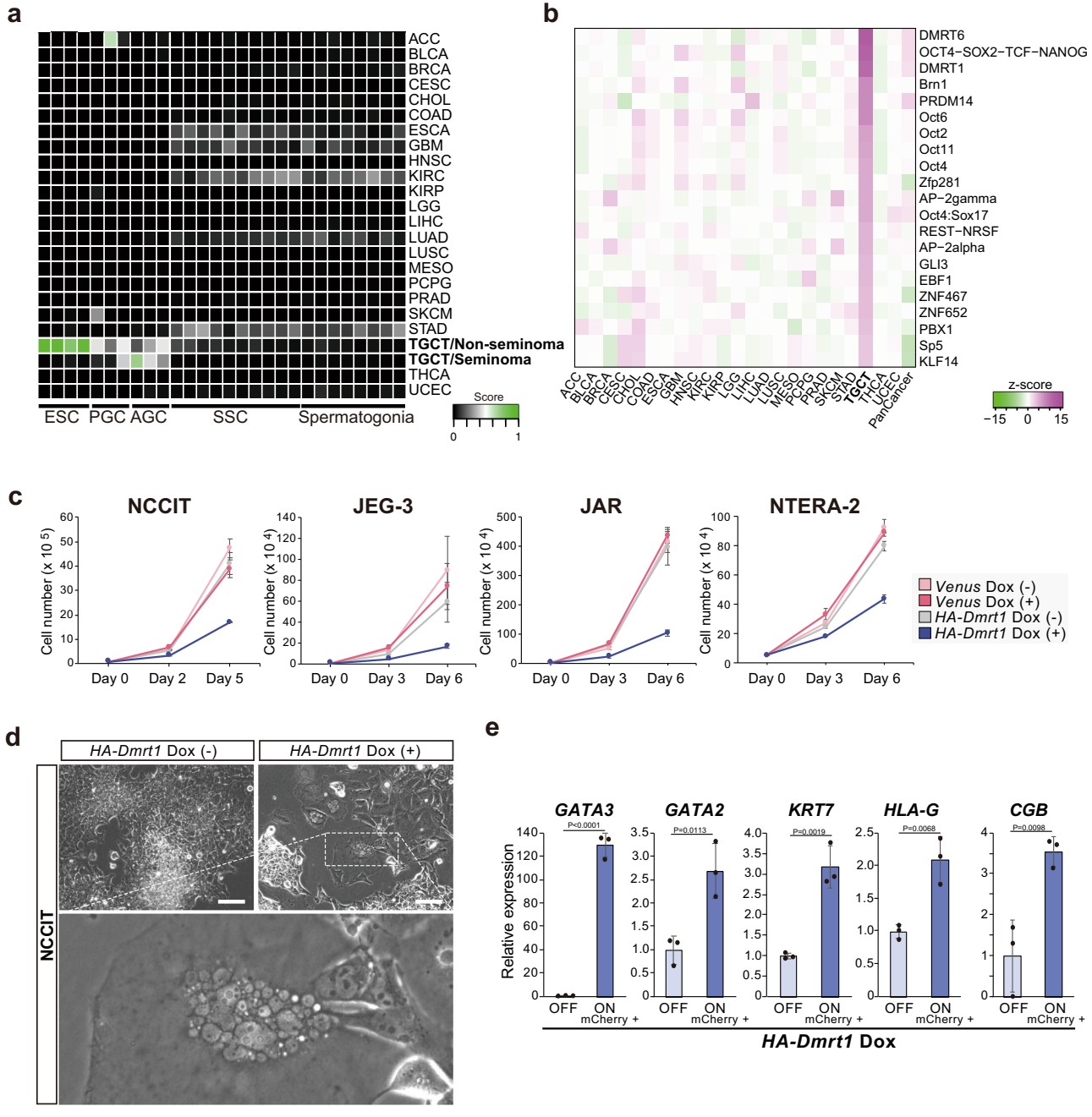

**Fig. 7 DMRT1-mediated reprogramming provides a therapeutic target for human germ cell tumors. a** Heatmap showing transcriptional correlation between human PSCs, germ-line cells, and 23 types of human cancer; analysis was performed using KeyGenes[77] (AGC, male advanced germ cell; SSC, spermatogonial stem cell). RNA-seq data of ESCs, PGCs, and AGCs were obtained from GSE63392[78], and those of SSCs and spermatogonia were obtained from GSE92280[79]. RNA-seq data from 23 types of human cancer were obtained from the Cancer Genome Atlas (TCGA) (ACC: adrenocortical carcinoma [$n = 79$]; BLCA: bladder urothelial carcinoma [$n = 408$]; BRCA: breast-invasive carcinoma [$n = 1092$]; CESC: cervical squamous cell carcinoma and endocervical adenocarcinoma [$n = 304$]; CHOL: cholangiocarcinoma [$n = 36$]; COAD: colon adenocarcinoma [$n = 456$]; ESCA: esophageal carcinoma [$n = 164$]; GBM: glioblastoma multiforme [$n = 166$]; HNSC, head and neck squamous cell carcinoma [$n = 501$]; KIRC: kidney renal clear-cell carcinoma [$n = 530$]; KIRP: kidney renal papillary cell carcinoma [$n = 289$]; LGG: brain lower-grade glioma [$n = 512$]; LIHC: liver hepatocellular carcinoma [$n = 371$]; LUAD: lung adenocarcinoma [$n = 515$]; LUSC: lung squamous cell carcinoma [$n = 501$]; MESO: mesothelioma [$n = 86$]; PCPG: pheochromocytoma and paraganglioma [$n = 179$]; PRAD: prostate adenocarcinoma [$n = 496$]; SKCM: skin cutaneous melanoma [$n = 468$]; STAD: stomach adenocarcinoma [$n = 380$]; TGCT/non-seminoma: testicular germ cell tumor/nonseminoma [$n = 68$]; TGCT/seminoma: testicular germ cell tumor/seminoma [$n = 66$]; THCA: thyroid carcinoma [$n = 502$]; UCEC: uterine corpus endometrial carcinoma [$n = 555$]). **b** Heatmap indicates robust Z-scores of proportions of open-chromatin regions specific to 23 types of cancer with consensus motifs for each of the indicated transcription factors (see details in "Methods" section). Transcription factors whose binding motifs are present in the highest proportions of TGCT cancer-specific open-chromatin regions are shown. Each cancer-type-specific peak set, determined by ATAC-seq, was obtained from TCGA in the NIH Genomic Data Commons portal[80] (ACC, $n = 9$; BLCA, $n = 10$; BRCA, $n = 74$; CESC, $n = 2$; CHOL, $n = 5$; COAD, $n = 38$; ESCA, $n = 18$; GBM, $n = 9$; HNSC, $n = 9$; KIRC, $n = 16$; KIRP, $n = 34$; LGG, $n = 13$; LIHC, $n = 17$; LUAD, $n = 22$; LUSC, $n = 16$; MESO, $n = 7$; PCPG, $n = 9$; PRAD, $n = 26$; SKCM, $n = 13$; STAD, $n = 21$; TGCT, $n = 9$; THCA, $n = 14$; UCEC, $n = 13$). PanCancer represents 562,709 reproducible ATAC-seq peaks obtained from 410 tumor samples[80]. **c** Cell proliferation assay of *Dmrt1*-induced human GCT cell lines. Data are presented as the means ± SD of three independent experiments. **d** Representative images of *Dmrt1*-induced NCCIT at Day 5. A magnified image of a multinuclear syncytiotrophoblast-like cell is shown in the lower panel. Scale bars: 100 μm. **e** qRT-PCR analyses for expression of trophoblast-related genes in *Dmrt1*-expressing NCCIT cells. mCherry-positive *Dmrt1*-expressing cells were isolated using FACS. Data are presented as the means ± SD of biological triplicates. The mean expression level of Dox OFF cells was set to 1 (*t*-test, two-sided).

---

GW-TOPO vector, transferred into the p*Rosa26*-DEST vector (Addgene #21189), and used as a targeting vector.

*Nanog P2A-CreERT2* targeting vector. A 3.8-kb fragment containing *P2A-CreERT2-pA-PGK-Bsd-pA* with 50-bp homology arms was established using KAPA HiFi HotStart ReadyMix. This fragment was inserted into the 3′ side of exon 4 of *Nanog* in a BAC (BACPAC Resources Center) using the Red/ET BAC-recombination system. The fragment of *Nanog-P2A-CreERT2-pA-PGK-Bsd-pA* sequence with 5.0-kb (5′ arm) and 1.5-kb (3′ arm) homology arms was retrieved and used as a targeting vector.

Dox-inducible *piggyBac* vectors. HA-*Dmrt1* cDNA was synthesized by a commercial service (Eurofins Genomics). HA-*Dmrt1* or *Venus* cDNA was cloned using KOD-Plus-Neo. This fragment was inserted into the pCR8-GW-TOPO vector and transferred into the PB-TAC-ERN vector[57].

pBS31 vectors for KH2 system. *Dmrt1* cDNA was synthesized (Eurofins genomics) and cloned using KOD-Plus-Neo. *Cdx2* was cloned from cDNA derived from TSCs using KOD-Plus-Neo. These fragments were inserted into the pCR8-GW-TOPO vector and transferred into the pBS31 vector.

**Cell culture**. ESCs and S-/D-iPSCs were cultured in ESC medium (Knockout DMEM [Gibco], 2mM L-glutamine [Nacalai Tesque], 100× Nonessential amino acids [Nacalai Tesque], 100 U/mL penicillin and 100 μg/mL streptomycin [Nacalai Tesque], 15% FBS [Gibco], 0.11 mM mercaptoethanol [Gibco], and 1000 U/mL human LIF [Wako]) on mitomycin-C-treated MEFs. Human cancer cell lines were cultured in DMEM (Nacalai Tesque) containing 2mM L-glutamine (Nacalai Tesque), 100× Nonessential amino acids (Nacalai Tesque), 100 U/mL penicillin and 100 μg/mL streptomycin (Nacalai Tesque), 10% FBS (Gibco), and 0.11 mM mercaptoethanol (Gibco).

**Establishment of ESCs**

*D-OSKM ESCs* (Rosa26-M2rtTA; Co1a1::tetO-OSKM-ires-mCherry; Rosa26::tetO-OSKM-ires-mCherry). The KH2 OSKM ESC line was described previously[11]. KH2 OSKM ESCs ($1 \times 10^7$ cells) were suspended in 700 μL of PBS containing linearized *Rosa26::tetO-OSKM-ires-mCherry* targeting vector (20 μg), and electroporation was performed on a Gene Pulser Xcell (Bio-Rad). After electroporation, $2.0 \times 10^6$ ESCs were placed over mitomycin-C-treated MEFs in a 6-cm dish and cultured at 37 °C in ESC medium. Twenty-four hours after electroporation, antibiotic selection was performed with the 15 μg/mL blasticidin S (Funakoshi). The surviving ESC colonies were picked and expanded to establish the D-OSKM ESC line. PCR-based genotyping was performed to confirm the correct targeting by homologous recombination.

*D-OSKM (Pax8 or Pdx1) ESCs* (Rosa26-LSL-rtTA3; Col1a1::tetO-OSKM-ires-mCherry; Rosa26::tetO-OSKM-ires-mCherry; BAC Pax8- or Pdx1-Cre). The *Rosa26-M2rtTA* allele in D-OSKM ESCs was replaced by retargeting with a linearized *Rosa26-LSL-rtTA3* targeting vector (10 μg). Twenty-four hours after

electroporation, antibiotic selection was performed with 350 μg/mL G418 (Nacalai Tesque). After removal of the blasticidin S selection cassette from the *Rosa26::tetO-OSKM-ires-mCherry* allele using a vector expressing Dre recombinase (pCAGGS-DreO), *Pax8-Cre* or *Pdx1-Cre* BAC was introduced by electroporation, and anti-biotic selection was performed with 15 μg/mL blasticidin S. The surviving ESC colonies were picked and expanded to establish the D-OSKM (*Pax8* or *Pdx1*) ESC line.

*Nanog reporter ESCs (*Nanog-P2A-CreERT2, Rosa26-LSL-Venus, Rosa26-M2rtTA, Col1a1::tetO-OSKM-ires-mCherry). The wild-type *Rosa26* allele in KH2 OSKM ESCs was replaced by a *Rosa26-LSL-Venus* allele with a linearized *Rosa26-LSL-Venus* targeting vector (20 μg). After antibiotic selection with 350 μg/mL G418, the *Nanog-P2A-CreERT2* allele was introduced with a linearized *Nanog-2A-CreERT2* targeting vector (20 μg). After antibiotic selection with 15 μg/mL blasticidin S, the surviving ESC colonies were picked and expanded to establish the *Nanog* reporter ESC line. PCR-based genotyping was performed to confirm the correct targeting by homologous recombination.

*Dox-inducible* Dmrt1 *or* Cdx2 *ESCs*. The Flip-in system was used to introduce the *tetO-HA-Dmrt1-ires-mCherry* or *tetO-Cdx2-ires-mCherry* allele into KH2 ESCs containing a *Rosa26-M2rtTA* allele[58]. After antibiotic selection with 150 μg/mL hygromycin B (Roche), surviving ESC colonies were picked and expanded to establish Dox-inducible *Dmrt1* or *Cdx2* ESC lines.

**GFP labeling of PSCs**. For GFP labeling of PSCs, a *piggyBac* (PB) transposon vector carrying *CAG-EGFP-ires-Neo* and a PB transposase plasmid (pCAG-PBase)[57] were transfected using Xfect mES cell transfection reagent (Clontech). After antibiotic selection with 350 μg/mL G418, GFP-expressing ESC colonies were picked and expanded to establish GFP-labeled ESC lines. Immunofluorescence signals were detected on a BZ-X710 fluorescence microscope (KEYENCE).

**Knockout using CRISPR–Cas9**. sgRNAs were designed using CRISPRdirect (http://crispr.dbcls.jp/). sgRNA sequences were GACTGGAATGGCTACGCACC for *Cdx2*, ATAATTTTGCGGCGGCCCGCA for *Tead4*, and GAGCGTGAGGAAC CTCCGTC for *Dmrt1*. Each sgRNA was ligated into pSpCas9(BB)-2A-Puro (PX459; Addgene #62988). Four micrograms of each plasmid was transfected into $1 \times 10^4$ D-OSKM (*Pdx1*) ESCs using Xfect mES cell transfection reagent. After antibiotic selection with 1 μg/mL puromycin, the surviving ESC colonies were picked into 96-well plates and expanded. For genotyping, genomic sequences of exon 1 of *Cdx2*, exon 2 of *Tead4*, and exon 1 of *Dmrt1* were PCR-amplified using GO Taq Green Master Mix (Promega).

**Establishment of MEFs**. After blastocyst injection of ESCs, chimeric embryos were harvested at E14.5 and minced with razors. MEFs were cultured in DMEM (Nacalai Tesque) containing 2 mM L-glutamine (Nacalai Tesque), 100× Non-essential amino acids (Nacalai Tesque), 100 U/mL penicillin and 100 μg/mL streptomycin (Nacalai Tesque), 10% FBS (Gibco), and 0.11 mM mercaptoethanol (Gibco). To purify ESC-derived MEFs, antibiotic selection was performed with

1 µg/mL puromycin (Sigma) for 1 week, and puromycin-resistant MEFs were used for further experiments.

**Reprogramming of MEFs.** MEFs (passage 3–5) were seeded in $7.5 \times 10^4$ cells/well in 6-well plates in ESC medium. The next day, Dox was added at a concentration of 2 µg/mL (Day 0). The medium was replaced every day. NANOG-expressing iPSC-like cell colonies were evaluated after Dox withdrawal for seven days following Dox treatment for 7, 14, or 21 days. For immunostaining of NANOG, the cells were washed with PBS, fixed with 2% paraformaldehyde for 15 min at room temperature, and blocked with blocking buffer (PBS containing 0.1% Triton X-100 [Sigma] and 3% BSA [Wako]) at room temperature for 30 min. Cells were incubated overnight at 4 °C with rabbit monoclonal anti-NANOG (Cell Signaling Technology, Cat#8822, dilution 1/500). The next day, cells were treated with secondary antibody conjugated with Alexa Fluor 647 (Invitrogen, Cat#A-21244, dilution 1/500) and DAPI (Invitrogen, Cat#D21490, dilution 1/750) in blocking buffer at room temperature for 1.5 h. Immunofluorescence signals were detected on a BZ-X710 fluorescence microscope.

**Derivation of PSCs from OSKM tumors.** Kidney and pancreatic tumors in S-/D-OSKM mice were carefully resected. Pieces of tumors (0.5–1 cm³) were minced with razors and digested in DMEM with 1 U/mL Collagenase P (Roche) for 20–30 min at 37 °C. Suspensions were filtered through 100-µm and 40-µm cell strainers (BD) and pelleted ($200 \times g$, 5 min, 4 °C). For kidney tumors, pellets were resuspended in 0.25% trypsin-EDTA (Nacalai Tesque), incubated at 37 °C for 10 min, and pelleted ($200 \times g$, 5 min, 4 °C). The pellets were resuspended in ESC medium, and the cells were counted on a TC10 automatic cell counter (Bio-Rad). Tumor cells were seeded in 6-cm dishes and cultured on feeder cells in ESC medium. The medium was replaced every day. After seven days, iPSC-like colonies with dome-shape morphologies were picked into 96-well plates and expanded.

**Establishment of subclones from a single PSC.** GFP-positive single cells were isolated on a FACS Aria-II (BD) and seeded in 96-well plates on feeder cells. After expansion, the cells were passaged into 24-well plates, followed by a second passage into 6-well plates, and then used for teratoma-formation assay.

**Establishment of *Dmrt1*-induced cells.** ESCs carrying Dox-inducible *Dmrt1* allele were seeded in 6-well plates on feeder cells at $2 \times 10^4$ cells/well in TSC medium (RPMI 1640 [Gibco], 100× Nonessential amino acids [Nacalai Tesque], 100× sodium pyruvate [Gibco], 100 U/mL penicillin and 100 µg/mL streptomycin [Nacalai Tesque], 20% FBS [Gibco], 0.11 mM mercaptoethanol and 1.5 µg/mL heparin [Sigma], and 37.5 ng/mL FGF4 [Sigma]) containing 2.0 µg/mL Dox. After four days, the cells were passaged into 0.2% gelatin-coated 6-cm dishes, followed by a second and third passage into 10 cm dishes. From the first passage onward, the cells were cultured under the MEF medium without Dox. Second and third passages were performed when the cells were approximately 80% confluent. For inoculation of *Dmrt1*-induced cells, $3 \times 10^6$ cells in 25 µL of PBS were injected into the testis of 6-week-old C57BL/6 N male mice (Japan CREA). Formed tumors were excised and evaluated at four weeks after the injection.

**Human cancer cell lines carrying Dox-inducible *HA-Dmrt1* and *Venus*.** Human cancer cell lines NCCIT, NTERA-2 a1.D1, JAR, JEG-3, and SKBR3 were obtained from ATCC (CRL-2073, CRL-1973, HTB-144, HTB-36, and HTB-20). A549 and HeLa were obtained from RIKEN (RBRC-RCB0098, RBRC-RCB0007). PANC-1 was obtained from DS pharma biomedical (EC87092802-F0). All samples used were tested for mycoplasma contamination. We confirmed that our cell lines are negative to mycoplasma contamination. The *piggyBac* transposon vector carrying *tetO-HA-Dmrt1/Venus-ires-mCherry-Ef1a-rtTA* and pCAG-PBase was transfected into human cancer cell lines using Lipofectamine 2000 (Thermo Fisher Scientific). After antibiotic selection with 350 µg/mL G418, the surviving cells were expanded and used for analyses.

**Cell proliferation assay.** *Dmrt1* KO ESCs were seeded in 6-well plates at $2 \times 10^4$ cells/well on feeder cells (Passage 1). Passage was performed every three days, and the cells were counted on a TC10 automated cell counter. Human cancer cell lines were seeded in 6-well plates (NCCIT: $1 \times 10^5$ cells/well; JAR: $2 \times 10^4$ cells/well; JEG-3: $1 \times 10^4$ cells/well; other cancer cell lines: $5 \times 10^4$ cells/well). Dox was added at a concentration of 1 µg/mL for NCCIT and JAR, and 2 µg/mL for JEG-3 and other cancer cell lines (Day 0). On the indicated days, cells were counted on a TC10 automatic cell counter. Experiments were conducted in biological triplicate.

**Embryoid body formation.** For EB differentiation, the cells were seeded in Nunclon Sphera Microplates (Thermo Fisher Scientific) at $3 \times 10^4$ cells/well in ESC medium without LIF.

**Alkaline phosphatase (AP) staining.** AP staining was performed using the AP Staining Kit II (STEMGENT).

**Fluorescence-activated cell sorting (FACS).** For preparation of in vitro samples, cells were washed with PBS and incubated in 0.25% trypsin-EDTA (Nacalai Tesque) for 5 min at 37 °C. For in vivo tumor samples, dissociation of tumor tissues was performed as described above. Cell pellets were resuspended in FACS buffer (PBS containing 4% BSA) and passed through a cell strainer. GFP- or mCherry-positive cells were sorted by FACS Aria-II or Aria-III (BD).

**Mice.** All animal experiments were approved by the Animal Experiment Committee at IMSUT and CiRA, and animal care was conducted in accordance with institutional guidelines. All mice were housed in a specific pathogen-free animal facility under a 12-h light/12-h dark cycle with food and water ad libitum.

**Generation of chimeric mice.** Eight-week-old ICR female mice (Japan SLC) received 7.5 U of serotropin (ASKA Animal Health) by intraperitoneal injection. Forty-eight hours after serotropin treatment, mice were injected with 7.5 U of gonadotropin (ASKA Pharmaceutical) and then mated with ICR male mice (Japan SLC). Two-cell fertilized eggs were collected by perfusion with mWM medium and maintained in KSOM medium to obtain blastocysts. After injection of six to ten ESCs, the injected blastocysts (22–26 blastocysts/mouse) were transplanted into the uterus of pseudopregnant ICR female mice (Japan SLC). A single ESC line for each genotype was used in this study.

**In vivo reprogramming.** S-OSKM/D-OSKM chimeric mice at 4–8 weeks of age received 2 mg/mL Dox (Sigma) in drinking water supplemented with 10 mg/mL sucrose (Nacalai Tesque). For the lower-dose Dox treatment, Dox was administered at a concentration of 0.13, 0.25, and 0.4 mg/mL for *Pax8-Cre* mice (kidney) and 0.1 mg/mL for *Pdx1-Cre* mice (pancreas). For in vivo reprogramming, mice received Dox-containing water for 7 or 14 days, followed by withdrawal for seven days. Mice exhibiting a highly morbid phenotype before the termination of the experiment were excluded from the analyses.

**Tamoxifen treatment.** Tamoxifen (Sigma) was dissolved in corn oil (Wako) at a concentration of 10 mg/mL. Mice were intraperitoneally treated with 2.0 mg of tamoxifen six times every other day.

**Teratoma-formation assay.** For teratoma formation assays, $2 \times 10^6$ cells in 200 µL of PBS were injected into the subcutaneous tissue of 6-week-old BALB/c-nu/nu immunodeficient female mice (Japan CREA). The resultant tumors were excised and analyzed three weeks after the injection. For tumorigenicity assays in *Dmrt1* KO ESCs, tumor volume was measured every four days starting at 14 days (Day 0) after injection until Day 16. Relative tumor volumes were calculated by dividing the measured tumor volume at Day 4, Day 8, Day 12, and Day 16 by the corresponding tumor volume at Day 0. For secondary tumor formation, D-OSKM tumors were minced and $1 \times 10^6$ cells were injected into the subcutaneous tissue of 6-week-old BALB/c-nu/nu immunodeficient female mice.

**Microinjection of PSCs into eight-cell embryos for chimera assay.** Eight to ten GFP-labeled PSCs were microinjected into the perivitelline space of 8-cell embryos using the PiezoXpert (Eppendorf) under an OLYMPUS IX71 microscope. Twenty-four to thirty-six hours after microinjection, localization of GFP-labeled cells was evaluated in blastocysts. The injected blastocysts were transplanted into the uterus of pseudopregnant ICR female mice (Japan SLC), and the contribution of GFP-labeled cells was examined in E13.5 concepti.

**Histological analysis, immunostaining, and immunofluorescence.** Dissected tissue samples were fixed in 4% PFA (Wako) overnight at room temperature. The fixed samples were embedded in paraffin using an STP120 spin tissue processor (Thermo Scientific) or HistoCore PEARL (Leica Biosystems). Sections were sliced to a thickness of 3–4 µm and stained with hematoxylin and eosin (HE). Serial sections were used for immunohistochemical analyses. The samples were soaked three times for 5 min each in xylene to remove paraffin, and three times for 5 min each in 100% ethanol to hydrophilize. After water washing for several minutes, the samples were soaked in epitope-retrieval buffer (Nichirei Bioscience) and microwaved at 100 W for 10 min. The samples were then soaked in PBS for several minutes and incubated with 200 µL of primary antibodies in PBS with 2% BSA (MP Biomedicals) at 4 °C overnight. The primary antibodies used were mouse monoclonal anti-OCT4 (BD, #611203, dilution 1/200), rabbit monoclonal anti-NANOG (Cell Signaling Technology, Cat#8822, dilution 1/1000), rabbit monoclonal anti-Ki67 (Abcam, #ab16667, dilution 1/200), rabbit polyclonal anti-2A peptide (Merck Millipore, #ABS31, dilution 1/250), rabbit monoclonal anti-DAZL (Abcam, #ab215718, dilution 1/200), rabbit monoclonal anti-CDX2 (Thermo Fisher Scientific, #MA5-14494, dilution 1/200), mouse monoclonal anti-AP-2γ (Tfap2c; Santa Cruz Biotechnology, #sc-12762, dilution 1/200), goat polyclonal anti-Placental lactogen I (PL-1; Santa Cruz Biotechnology, #sc-34713, dilution 1/100), and rabbit monoclonal anti-GFP (Abcam, #ab183734, dilution 1/200). Sections were incubated with the HRP-conjugated secondary antibodies (Nichirei, Histofine) at room temperature for 30 min, and chromogen development was performed using DAB (Nichirei). The stained slides were counterstained with Meyer hematoxylin. For

immunofluorescence, after incubation with primary antibodies, sections were stained for 90 min at room temperature in PBS with 2% BSA with secondary antibody conjugated with fluorescent protein: CF488A anti-mouse IgG (Biotium, #20014, dilution 1/500), CF555 anti-rabbit IgG (Biotium, #20038, dilution 1/500), and DAPI (Invitrogen, #D21490, dilution 1/750). After two more 5-min washes in PBS, the sections were mounted using ProLong Gold antifade reagent (Invitrogen) and evaluated with a confocal laser scanning microscope LSM700 or LSM710 (ZEISS).

**Alcian blue staining**. Deparaffinized slides were pretreated with 3% acetate for 5 min. Sections were stained with Alcian blue 8GX solution (Merck Millipore) for 30 min, and then with Nuclear Fast Red solution (Merck Millipore) for 5 min.

**Immunocytochemistry**. Cultured cells were fixed with 2% paraformaldehyde for 15 min at room temperature and treated with blocking buffer (PBS containing 0.1% Triton X-100 [Sigma] and 3% BSA [Wako]) at room temperature for 30 min. Cells were stained overnight at 4 °C with rabbit monoclonal anti-CDX2 (Thermo Fisher Scientific, #MA5-14494, dilution 1/200). The next day, cells were stained in blocking buffer at room temperature for 1.5 h with secondary antibody conjugated with Alexa Fluor 647 (Invitrogen, #A-21244, dilution 1/500) and DAPI (Invitrogen, #D21490, dilution 1/750). Immunofluorescence signals were detected on a BZ-X710 fluorescence microscope.

**Quantification for histological analysis**. For quantification of histological features of tumors, tumors on HE-stained sections and immunostained sections were randomly photographed at 100× magnification. Four or five images from each tumor were processed with ImageJ software (NIH) to evaluate the region of interest. The histological region containing more than three trophoblast giant cells (TGCs) in a high-power field (×400) was defined as a TGC cluster. To calculate the ratio of positive cell area, the positive area was divided by the total tumor area in each image. For assessment of Ki67-positive cells, the OCT4-positive cell area in each section was randomly imaged at 200× magnification. Cells with positive nuclei were counted in five images. To calculate the ratio of Ki67-positive cells, the number of positive nuclei was divided by the total number of nuclei in each image. For quantification of tumor area, coronal sections of the pancreas or kidney were imaged on a BZ-X710 microscope, and a single image covering the total area of the organ was obtained by stitching these images together. The single image was processed with the ImageJ software. To calculate the tumor area, the histologically undifferentiated cell area was divided by the total area of the organ. These results were evaluated using the GraphPad Prism 7 software.

**Western blot analysis**. Cultured cells were harvested in 500 μL of RIPA lysis buffer (10 mM Tris-HCl [pH 8.0], 150 mM HCl, 1% Triton, 1% DOC, and 0.1% SDS) containing 0.5% protease inhibitor, 1% DTT, and 100× phosphatase inhibitor (Nacalai Tesque) on ice. Twenty micrograms of denatured protein was loaded per lane of a 10% SDS/PAGE gel; after electrophoresis, the proteins were transferred onto Amersham Hybond-P PVDF Membrane (GE HealthCare). Membranes were incubated with primary antibodies in blocking buffer (4% skim milk in TBST) overnight at 4 °C. The primary antibodies used were mouse monoclonal anti-OCT4 (BD, #611203, dilution 1/500), mouse monoclonal anti-SOX2 (Merck Millipore, #MAB4343, dilution 1/100), rabbit polyclonal anti-KLF4 (Thermo Scientific, #PA5-27440, dilution 1/300), rabbit monoclonal anti-c-MYC (Abcam, #ab32072, dilution 1/100), mouse monoclonal anti-TEAD4 (Abcam, #ab58310, dilution 1/100), mouse monoclonal anti-DMRT1 (Santa Cruz Biotechnology, #sc-377167, dilution 1/100), and mouse monoclonal anti-GAPDH (Invitrogen, #AM4300, dilution 1/1000). The membrane was washed in TBST and incubated with secondary antibodies in blocking buffer for 1 h at room temperature. The secondary antibodies used were sheep anti-mouse IgG (GE HealthCare, #NA931, dilution 1/1000) and donkey anti-rabbit IgG (GE HealthCare, #NA934, dilution 1/1000). The membrane was incubated with Pierce ECL plus Western Blotting Substrate (Thermo Scientific) for visualization. The bands were detected on an Amersham Imager 680 (GE HealthCare).

**Genomic DNA extraction**. Feeder cells were removed before DNA extraction from in vitro PSC samples. For in vivo samples, freshly collected tissues were frozen in liquid nitrogen and ground into powder using a mortar. Genomic DNA was extracted using the PureLink Kit (Invitrogen), according to the manufacturer's protocol, and quantified on a NanoDrop 2000c (Thermo Fisher Scientific) and Qubit (Thermo Fisher Scientific).

**Bisulfite sequencing**. Two-hundred nanograms of DNA was bisulfite-treated using EZ DNA Methylation-Gold Kit™ (ZYMO RESEARCH). PCR was performed with Ex Taq HS or LA Taq HS (TaKaRa) with primers listed in Table S1. PCR products were cloned into the pCR4-TOPO vector. After transformation into DH5α, colony PCR was performed using GO Taq Green Master Mix (Promega). After cleanup of PCR products by exonuclease I (New England Biolabs) and shrimp AP (TaKaRa), PCR products were sequenced with M13 reverse primer. Sequencing results were evaluated using the QUMA software (RIKEN).

**Library preparation for whole-genome bisulfite sequencing (WGBS)**. DNA (500 ng, as determined by Qubit) was fragmented by sonication (Covaris) and ligated with methylated adapters using the TruSeq Nano DNA Library Prep Kit and TruSeq ChIP Library Preparation Kit (Illumina). Subsequently, DNA was bisulfite-treated using the EZ DNA Methylation-Gold Kit. Final library amplification was performed using *Pfu* Turbo Cx (Agilent Technologies). Sequencing libraries were assessed on a Bioanalyzer (Agilent Technologies) and quantified using the KAPA Library Quantification Kit (KAPA BIOSYSTEMS). The libraries were then sequenced as paired-end reads (read 1, 109 bp; read 2, 108 bp) on a HiSeq 2500 (Illumina) in rapid mode.

**Library preparation for target-captured bisulfite sequencing (MethylC-seq)**. DNA (1 μg, as determined by Qubit) was fragmented by sonication. Subsequently, library preparation was performed using the SureSelect Mouse Methyl-Seq Reagent Kit (Agilent Technologies). The Methyl-Seq Kit could enrich 109-Mb mouse genomic regions including CpG islands, Gencode promoters, DNase I-hypersensitive sites, and tissue-specific DMRs. DNA was bisulfite-treated using the EZ DNA Methylation-Gold Kit. Sequencing libraries were assessed on a Bioanalyzer and quantified using the KAPA Library Quantification Kit. The libraries were then sequenced as paired-end reads (read 1, 109 bp; read 2, 108 bp) on a HiSeq 2500 (Illumina) in rapid mode.

**DNA methylation data analysis**. For WGBS and MethylC-seq analyses, bases with low-quality scores and the adapters in all sequenced reads were trimmed with cutadapt-1.18[59]. Trimmed reads were mapped to mm10 mouse reference genomes using the Bismark software (version 0.20.1)[60] with bowtie2 (version 2.3.5)[61]. Methylated cytosines were extracted from reads using the Bismark methylation extractor with the following options: --ignore 10 --ignore_r2 10 --ignore_3prime 5 --ignore_3prime_r2 5. For MethylC-seq analysis, because probe sequences of the SureSelect were designed for the original top strand, only the original bottom (OB) data were used for this analysis. Genomic locations for each genetic element, including mouse ICRs, were described previously[62,63].

**RNA preparation**. For RNA extraction from in vitro PSC samples, cells were attached onto gelatin-coated dishes for 30 min to remove feeder cells. For in vivo samples, freshly collected tissues were frozen in liquid nitrogen and ground into powder using a mortar. Total RNA was isolated using the RNeasy Plus Micro or Mini Kit (QIAGEN). RNA was quantified on a NanoDrop 2000c and Qubit.

**cDNA synthesis and qPCR analysis**. Five-hundred nanograms of RNA was reverse-transcribed into cDNA by using the PrimeScript RT Reagent Kit (Takara Bio). The quantitative real-time PCR analysis was performed using the GoTaq qPCR Master Mix and CXR Reference Dye (Promega) on a StepOnePlus Real-Time PCR system (Thermo Fisher Scientific). The primers used are shown in Table S1. Transcript levels were normalized against the corresponding level of *Actb*. Experiments were performed in biological triplicate.

**Library preparation for RNA sequencing**. RNA (in vitro and in vivo samples; 200 ng, FACS-sorted mCherry-positive tumor cells; 50 ng) was prepared for library construction. High-quality RNA (RNA Integrity Number value ≥8, as determined by Bioanalyzer) was used for library preparation. RNA-seq libraries were generated using the TruSeq Stranded mRNA LT sample prep kit (Illumina) or NEBNext Ultra II Directional RNA Library Prep kit for Illumina (NEB). The number of PCR cycles was minimized to avoid skewing the representation of the libraries. RNA-seq libraries were sequenced on a NextSeq500 (Illumina) as 86-bp single reads.

**RNA-seq data analyses**. After trimming of adapter sequences and low-quality bases with cutadapt-1.18[59], sequenced reads were mapped to the mouse reference genome (mm10), and expression levels were calculated as TPM (transcripts per million) using RSEM 1.3.1[64] with the STAR (version 2.6.0c)[65] aligner and the GENCODE version M23 annotation gtf file[66].

**Library preparation for ATAC-seq analyses**. ATAC-seq libraries were prepared as previously described with some modifications[67]. Cells were lysed with a lysis buffer (10 mM Tris-HCl, pH 7.4, 3 mM MgCl₂, 10 mM NaCl, and 0.1% IGEPAL CA-630). Nuclei were suspended in the transposase reaction mix (25 μL of 2X TD Buffer [Illumina], 2.5 μL of TD Enzyme 1 [Illumina], and 22.5 μL of nuclease-free water) and incubated for 30 min at 37 °C. Fragmented DNA was then purified using the MinElute PCR purification kit (QIAGEN). After that, fragmented DNA was amplified by PCR to obtain ATAC-seq libraries. The resultant libraries were purified using the MinElute PCR purification kit and quantified using the KAPA Library Quantification Kit for Illumina (KAPA Biosystems). The libraries were subjected to paired-end sequencing (50 bp) on a HiSeq 1500. Total numbers of sequenced reads and the mapping efficiencies are shown in Supplementary Fig. 6a.

**ATAC-seq data analyses**. Adapter sequences in reads were trimmed using Trim Galore version 0.3.7. Trimmed reads were aligned at the mm10 genome build using

bowtie version 0.12.7 with default parameters[68]. Duplicated reads were removed using samtools version 0.1.18. MACS2 bdgdiff version 2.1.1 was used to identify loci with higher or lower accessibility in D-iPSCs with the parameters -g 100 -l 200[69]. Motif analyses were performed using HOMER version 4.8.3 with the parameter --mask[70]. The Integrative Genomics Viewer (IGV)[71] was used to visualize ATAC-seq peaks.

**ATAC-qPCR.** ATAC-qPCR was performed as described previously with minor modifications[72]. Briefly, ATAC-seq libraries in ESCs and D-iPSCs were generated according to the method described in the ATAC-seq section. Then, 1 ng of the ATAC-seq libraries was used in each qPCR reaction as a template. ATAC-qPCR values were normalized to Rpl2[72]. Primer pairs were designed using ATACPrimerTool[72], which are indicated in Table S1.

**Statistics and reproducibility.** All statistical parameters including sample size, statistical comparison test, and exact $p$-value, are described in figure or figure legends. Statistical analyses were performed using Prism 8 software (GraphPad). Data are presented as the means ± SD. The reproducibility of representative images was confirmed in a minimum of three biologically independent samples.

**Reporting summary.** Further information on research design is available in the Nature Research Reporting Summary linked to this article.

## Data availability

All data generated by RNA-seq, ATAC-seq, WGBS, and MethylC-seq in this study have been deposited in the Gene Expression Omnibus (GEO) under accession number GSE149499. All other relevant data supporting the key findings of this study are available within the article and its Supplementary Information files or from the corresponding authors upon reasonable request. The source data underlying Fig. 1c and Supplementary Fig. 1f, 3c and 8a are provided as a Source Data file. A reporting summary for this article is available as a Supplementary Information file. Source data are provided with this paper.

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

## Acknowledgements

We are grateful to R. Sakamoto, M. Kikuchi, M. Baba, S. Sakurai, and D. Seki for technical assistance. Yasuhiro Y. was supported in part by the Cancer Research Grant P-CREATE (JP18cm0106203h0003), Japan Agency for Medical Research and Development (AMED): AMED-CREST (JP19gm1110004h9903), AMED-SICORP (JP17jm0210039), AMED-JSPS KAKENHI (18H04026, 20H05384); the Princess Takamatsu Cancer Research Fund; the Takeda Science Foundation; the Mochida Foundation; and the Naito Foundation. T. Y. was supported by the Core Center for iPS Cell Research, Research Center Network for Realization of Regenerative Medicine, AMED. T. Y. was supported by AMED-CREST (19gm1110004s0203), JSPS KAKENHI 19H03418, and the iPS Cell Research Fund. The Institute for the Advanced Study of Human Biology (ASHBi) is supported by the World Premier International Research Center Initiative (WPI), MEXT, Japan. Y.S. and Masaki.K. were supported in part by SICORP (15652288), AMED.

## Author contributions

J.T. and Yasuhiro Y. designed and conceived the study and wrote the paper. J.T., H.S., Yosuke Y., and K.O. generated vectors and cell lines, performed experiments. K.W. provided vectors. A.T. and M.O. performed blastocyst injections. S.O and T.U. generated WGBS, Methyl-seq, and RNA-seq libraries. K.M. analyzed FACS data. S.I.N. and M.S. provided PGCLCs. S.Y. and T.U. analyzed clinical samples. Mio.K., Masaki.K., K.F, S.O., Y.S., and T.Y. analyzed WGBS, Methyl-seq, and RNA-seq data.

## Competing interests

The authors declare no competing interests.

## Additional information

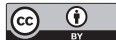

