## [Peer Review File · Nature Communications]

Reviewers' Comments:

Reviewer #1:

Remarks to the Author:

Taguchi et al generated a new mouse model that allows in vivo reprogramming with high levels of OSKM. They found that in vivo reprogramming leads to the generation and propagation of PGC-like cells. In addition, reprogramming with higher levels of OSKM increases trophectoderm differentiation within OSKM-derived tumors. They found *Dmrt1*, a transcription factor expressed in gonadal cells, as a promoter of in vivo reprogramming.

This is a nice piece of data that, on one hand, strengthens the concept of the extended potential of in vivo-generated PSC (already published but still unappreciated), and on the other hand provides new identities and regulators involved in in vivo reprogramming. Besides, the authors display an impressive amount of data, with state-of-the-art technologies. Having said that, I would like to mention some concerns that may help the authors to improve the paper:

1. Authors suggest that in vivo reprogramming implies a linear sequence of events: differentiated cells \rightarrow PGC-like cells \rightarrow Trophectodermal giant cells (TGCs). In other words, that PGC-like cells are an intermediate state, which have the potential to derive TGCs. Given that authors don't perform lineage tracing experiments with *DAZL* (or another marker of PGC-like cells) to demonstrate that TGCs are derived from PGCs, there is an alternative explanation: D-OSKM drives the generation of a mixture of independent cell populations: PGC-like cells (*Oct4+*/*DAZL+*), iPSC (*Oct4+*/*DAZL-*), and TGCs. Why do authors exclude this scenario, which in principle seems more plausible than the "strange" conversion (from a developmental point of view) of PGCs into TGCs? This alternative should be contemplated and discussed.

2. In relation to the above point, the scheme in Fig 6j is very confusing. Why the arrow towards PSCs say in vitro? Why *DMRT1* is acting at the level of GCT-like cells and not at PSCs to induce Trophectoderm differentiation?

3. It is not clear to me that D-OSKM induction drives the development of tumors that resemble germ cell tumors (GCT), and more specifically Type II non-seminomatous GCTs. Alternatively, D-OSKM tumors could be described as undifferentiated teratomas with a miscellanea of cells, some with PGC features, others with TGC features. It would be helpful to include some comparison (i.e. by transcriptomic profiles, markers) between ESC-derived teratomas, S-OSKM tumors, D-OSKM tumors and GCTs.

4. Page 14, line 352: Authors mention that D-iPSC are the only cells tested that have bidirectional potential, but in Fig 4c they show (in line with previous studies) that S-iPSC and ESCs also form tumors with TGCs. What is unique to D-iPSCs that is completely absent in S-iPSCs? The differences are quantitative, not qualitative.

5. Figure 2f,g: It would be informative to compare with the ICRs in S-OSKM tumors. Loss of imprinting could be a general phenomenon of in vivo reprogramming.

6. Following the same line, in Figure 4i it would be interesting to demonstrate the maintenance (or not) of ICR in S-iPSCs.

7. EBs are normally the first differentiation step to generate PGCs in vitro. Surprisingly, in this favorable context, D-iPSCs do not activate any features of PGCs (Extended Data Figure 4h). Isn't this surprising? It seems that the PGC features are only generated in vivo.

8. What happens if you monitor D-OSKM mice at longer time points? Do tumors become differentiated teratomas? This is an important point because GCTs do not become differentiated teratomas.

9. Page 16- *Dmrt1* deficiency decreases in vivo reprogramming efficiency (Fig 6b,c), so it is not surprising that also decreases trophectoderm differentiation. Is *Dmrt1* simply increasing in vivo reprogramming efficiency?

10. Page 19 Line 482: Authors state that D-iPSC exhibit a global transcriptional profile distinct from previously reported PSCs with similar differentiation potential. Where do they show these differences? Isn't this in conflict with the data shown in Extended Data Fig. 5b?

Minor points:

11. The abstract is very confusing. I had to read the paper to understand the abstract.

12. Page 11, line 271: Eomes is also detectable in in vitro reprogramming.

13. Authors should demonstrate that they have efficiently abolished the expression of CDX2 by CRISPR (as they do for Tead4 KO in extended Fig 3D).

14. Is it not clear why the pancreas and the kidney were chosen for reprogramming on the first place. Besides, kidney D-OSKM tumor and pancreas D-OSKM are used indistinctly with no clear criteria

15. Page 9, lines 209-212: These sentences are not very clear. I don't understand the authors' interpretation of sc-RNA-seq analysis. Please, clarify the message.

16. There are many mistakes in the numbering of the figures that difficult the reading and understanding of the manuscript (i.e. in line 217 Fig 2g should be 2f, in line 219 Fig 2h,i should be Fig 2 g,h, in line 221 Fig 2h should be 2g, in line 366, Fig 6d should be extended Fig 6d, line 403 Fig 6d shouldn't be 6b, 6h?).

17. Page 16, line 403: I guess authors wanted to say "However, Dmrt1-KO D-OSKM tumors still contained clusters of TGCs", and the referenced figure should be 6h.

18. Page 9, line 203-204; Page 12 line 284: Some references should be added here.

19. Line 230 is missing the abbreviation of trophoblast giant cells (TGCs).

Reviewer #2:

Remarks to the Author:

In this study, the authors found that in vivo expression of OSKM caused increased expression of PGC-related genes and provoked the genome-wide ICR demethylation in adult somatic cells. RNA-seq and methylation sequencing also confirmed this discovery. This reprogramming drove the development of a subgroup of human GCT-like cancers that exhibited trophoblast differentiation. Furthermore, the authors identified DMRT1 participated in reprogramming and associated with trophoblast competence using ATAC-seq technology. In general, this work is well organized and systematic, serving as a good reference for implications for GCT development and the acquisition of totipotency-like features by somatic cells. The sequencing data analysis is weak. I have a few comments:

1. In Fig 5a, ATAC-seq results indicated that DMRT1 was involved in reprogramming, however there were only 270 D-iPSC-specific peaks. The authors should change the method to identify sample type-specific peaks, because in Fig 1e, RNA-seq results suggested that there were great differences between D-OSKM and S-OSKM in gene expression. The authors should provide basic information about the ATAC-seq data, such as: how many replicates were used? how was the data quality? how many peaks in total were identified? How the differential peaks were evaluated? and the thresholds were used to indicate the level of significance. Without this information, it is difficult convince the readers. The method used to calculate the motif enrichment score in Fig 7d does not consider the number of TF binding sites in the whole genome, and therefore is problematic.

2. The authors claimed that they found a small subset of D-OSKM tumor cells through single-cell RNA-seq (Fig 2d) and showed the expressions of several PGC-related genes (Extended Data Fig 2b), however it seems that none of the D-OSKM tumors subtype cells expressed these four genes except for PGC cell subpopulation in Extended Data Fig 2b. Which group is the small subset of D-

OSKM tumor cells that author claimed? In addition to these four genes, what is the overall expression level of PGC-related genes in these tumors? How many cell subtypes were found through scRNA-seq, and what are they? Is cell clustering in Fig 2e over fitting, since there are many cell clusters contain very few number of cells? What are the parameters used to identify these cell subtypes, and how do the authors make sure the clustering is right? Does it mean that even though D-OSKM tumors express PGC-related genes, they were still different in transcriptome level? Fig. 2e and Extended Data Fig 2b should be combined and better labeling should be provided, otherwise readers won't understand which cell type is which.

3. In line 276, the authors described "lack the evidence of differentiation into three germ layers". Why not test the capacity to form teratomas of the minced D-OSKM kidney tumors directly?

4. The authors mentioned that DUX was responsible for the unique transcriptional signature in the 2C-like state, but in this model, the expression of DUX is very low. So, what is the difference between these two pathways? The author should compare the transcriptome difference between the two pathways.

5. The authors mentioned that D-OSKM tumors resembled human GCTs in the article, but the authors need to compare their differences at transcriptome level to prove D-OSKM tumors can be used as a model for GCT development.

6. The color bar of Fig 7a and Extended Data Fig 5b have no color gradient.

7. In Fig 2f and Fig 4i, the authors need to provide quantitative values to measure the level of methylation in different samples. No description about the coordinates will make reader very confused.

8. In lines 212 and 217, the figure label is incorrect, "However, these tumor cells were identified as a distinct population from PGCs and ESCs (Fig. 2e, f)", it should be "However, these tumor cells were identified as a distinct population from PGCs and ESCs (Fig. 2e)"

9. The author showed induction of OSKM in the kidney and pancreas drove in vivo reprogramming and propagation of GCT-like tumor cells, why not try to induction of OSKM in germ cells?

Reviewer #3:

Remarks to the Author:

The study described in this manuscript is a straightforward and significant extension of a series of preceding research performed by the group of Authors on in vivo carcinogenesis of mouse models that transiently express the OSKM reprogramming factors under control of exposure to doxycycline. In the current study, tissue-specific, doxycycline-inducible expression of large amounts of the OSKM reprogramming transcription factors in mouse kidney or pancreas in vivo resulted in generation of highly malignant cells (D-OSKM tumor cells), which express some of the pluripotency marker genes and show several signs of totipotency. Authors provided a series of persuasive evidence that DMRT1 plays critical roles in the observed gain of totipotency.

I think that the presented research has significant scientific merits, and it may be worth publishing from Nature Communications, pending revisions addressing the following concerns:

Number 1: Authors argue that INCREASED LEVELS of OSKM expression is the cause of the phenomenon. However, their own data indicate that stoichiometry of expression of the four factors is significantly different in the D-OSKM MEFs from that observed with the previously reported, S-OSKM MEFs. Authors are requested to address possible contributions of the IMBALANCED expression of the OSKM factors to the observed phenomena.

Number 2: Reproducibility of the experiments GENERATING the D-OSKM ESCs and MEFs is not clearly documented. How many independent D-OSKM ESC clones were isolated in the current study? Do they show the same, or at least similar, phenotypes in vitro and in vivo? It is also

important to describe how the individual D-OSKM ESC clones were characterized after electroporation of the linearized target vector in sufficient details. I am assuming that both S-OSKM and D-OSKM ESCs were genetically engineered by using the targeted knock-in technique at the Col1a1 and Rosa26 loci, but procedures of screenings and/or characterization is not described in the manuscript. Since electroporation of linearized plasmids may also generate insertions of plasmid concatemers at non-targeted sites in the genomic DNA, evaluation of such off-target insertions is often critical for studies on carcinogenesis using cell culture models.

Number 3: Since the D-OSKM is a mouse model, I am a little bit surprised by the total lack of mentions to mouse embryonal carcinoma cell culture models in the manuscript whereas Authors attempt to link the biological characteristics of the D-OSKM tumor cells to human embryonal carcinoma. The phenomenon that mouse embryonal carcinoma cells, which are highly malignant, are capable of contributing to normal tissue development in mice when injected into mouse embryos is already known (e.g., See, Astigiano et al. 2005, "Fate of embryonal carcinoma cells injected into postimplantation mouse embryos" *Differentiation* 73:484-490). Roles of *Dmrt1* in teratoma incidence in 129 mice as a repressor of pluripotency are also documented (See, a review article of Bustamante-Marin, Garness, and Capel 2013, "Testicular teratomas: an intersection of pluripotency, differentiation, and cancer biology" *Int. J. Dev. Biol* 57:201-210). To me, the D-OSKM system seems like an excellent model of MOUSE embryonal carcinomas, and its relevance to human embryonal carcinoma sounds indirect. Authors are requested to address why they think their model is more closely linked to human embryonal carcinoma than the same type of tumors or tumor cells in mice. In addition, I urge Authors to specify the human tumor types a little bit more carefully. It would be fair to suggest the resemblance of their D-OSKM model to human embryonal carcinomas based on their presented data, but I see multiple occasions that they attempted to extend their claim to Type II GCTs (including both seminomatous and non-seminomatous tumors) or even generally GCTs.

[Specific Comments]

Lines 137-140. The sentence, "The results indicate that different levels of OSKM expression resulted in a substantial difference in global transcriptional signatures, which presumably contributed to the differences in the efficiency of invitro iPSC derivation following withdrawal of OSKM expression," has the following three concerns:

Number 1: Authors stated in Lines 74-76, "Notably, the details of the reprogramming process depends on cell-intrinsic factors such as cell type, as well as the LEVELS AND STOICHIOMETRIES of reprogramming factors." Figure 1b (RT-qPCR) shows that mRNA expression of the reprogramming factors differs remarkably between the S-OSKM and D-OSKM MEFs in not only their LEVELS but also STOICHIOMETRIES. However, the sentence of Lines 137-140 mentions only the different LEVELS. Authors are requested to explain why they think that only the increased levels – not their apparent imbalance – of the reprogramming factor expression affected the transcriptomes.

Number 2: The basis of the claim that the observed substantial differences in global transcriptional signature likely contribute to the reprogramming efficiency needs more explanations. Because the PCA plot (Figure 1e) does not tell the degree of transcriptomal impact (number of affected genes of magnitudes of changes in expression), the claimed "substantial differences in global transcriptional signature" is not easily evaluated. Even if expression of a number of genes differ significantly between the S- and D-OSKM MEFs, whether such differential expression affects the reprogramming efficiency would need further explanations.

Number 3: In Figures 1b, 1d, and 1e, I can see three dots for each bar. Are these dots represent three independent clones of S- or D-OSKM? Alternatively, are these data show outcomes of three independent experiments performed using only a single clone of S- or D-OSKM? Authors are requested to clarify the number of independent S/D-OSKM clones involved in these figures and discuss the experimental reproducibility.

Lines 161-164. The sentence claims that D-OSKM reprogrammable mice developed tumors in kidney and pancreas with nearly complete penetrance. Authors are requested to present numbers

of mice and tumor incidence for S- and D-OSKM mice.

Lines 185 and 189. See, comment on Lines 137-140, Number 1.

Lines 191-193. The claim, "We found that D-OSKM tumors consisted of OCT4-positive proliferating cells [that] also expressed NANOG, INDICATING that D-OSKM tumors RESEMBLED human GCTs," sounds too strong based solely on mere expression of two pluripotency marker genes. Note that further evidence linking the D-OSKM tumor model to human embryonal carcinoma cells are described later in the manuscript. It is acceptable to state here, "We found that D-OSKM tumors consisted of OCT4-positive proliferating cells also expressed NANOG. Expression of these pluripotency markers is also known as characteristics of several subtypes of human germ cell tumors such as embryonic carcinomas or seminomas."

Lines 193-199. The presented pathological characteristics of D-OSKM tumors (invasive growth and central necrosis) are common features of highly malignant tumors. Since these are not specific characteristics of embryonic carcinoma, the last sentence, "Together, these findings show that ... resulted in development of cancers that shared characteristics with human malignant type II non-seminomatous GCTs," sounds too strong. It is acceptable to state, "... resulted in development of cancers with strongly malignant characteristics."

Lines 212, 217, and 219. I think the reference to figures in these lines are incorrect. Line 212: (Fig. 2e,f) -> (Fig. 2e). Line 217: (Fig. 2g) -> (Fig. 2f). Line 219: (Fig. 2h,I) -> (Fig. 2g, h). Note that there is no Figure 2I.

Lines 213-224, Fig. 2f,g,h, Extended Data Fig. 2c,d. The observed ICR-specific strong demethylation in the D-OSKM tumors is interesting. If tumor RNA samples are still available, Authors should examine mRNA expression of the imprinting genes whose ICR is strongly demethylated. Since Authors have RNA-seq data of S- and D-OSKM MEFs in the course of in vitro reprogramming towards PSCs (Fig. 1e), it would be appreciated if Authors examine expression of the imprinting gene mRNA transcripts.

Lines 241-245. Authors claim that the presence of cells reminiscent of trophoblastic giant cells and their precursors in the D-OSKM tumors provides additional evidence that D-OSKM tumors share properties with human GCTs. I would say that the D-OSKM tumors show signs of totipotency, which is a shared feature with human embryonal carcinoma.

Lines 246-252. The presented lineage tracing data demonstrate that Nanog-expressing cells give rise to cells resembling the trophoblast giant cells in the S-OSKM tumors. Although technical limitations may prevent lineage tracing experiments directly using the D-OSKM tumors, Authors are requested to carefully discuss whether the data obtained from the S-OSKM model can be adequately extrapolated to biology of the D-OSKM cells.

Lines 260-261 and Fig. 3i, j, Extended Data Fig. 3e. Areas of TGC clusters should be evaluated using PL-1 immunohistochemistry. The text referring to Fig. 3c (Lines 229-232) suggest that area of TGC cluster was evaluated by PL-1 immunohistochemistry, but neither figure legend nor Method (Lines 782-797) clearly states so. H&E staining images presented as Fig. 3i are not sufficiently informative to identify areas occupied by TGC clusters. If PL-1 staining was not involved in calculating areas of TGC clusters presented in either Fig. 3c or Fig. 3j, strong justification for the use of plain H&E images is required. This comment is also applicable to Fig. 4c, Extended Data Fig. 4e, and Fig. 5i.

Lines 266-271. Authors observed the increased expression of PGC- and trophoblast-related genes only during the in vivo reprogramming process. Authors are requested to discuss whether the in vitro cell culture conditions during reprogramming are adequate to support survival of PGC-like or TGC-like cells derived from the D-OSKM ESCs. Even if such cells could actually be generated in both the in vitro and in vivo reprogramming conditions, these cells may not be able to survive and/or proliferate in the cell culture conditions optimized for reprogramming.

Line 366. I think the referenced figure (Fig. 6d) is incorrect. It should be (Extended Data Fig. 6d).

Line 403. Authors claim that D-OSKM tumors still contained clusters of trophoblast giant cells (Fig. 6b), but is hard to identify the claimed trophoblastic giant cells in the image presented as Fig. 6b.

Line 411. Authors state that seminomas may represent precursors of non-seminomas. Although this possibility has been proposed, it is not well-demonstrated yet. Currently accepted notion is that both seminomas and embryonal carcinomas are derived directly from GCNIS and that embryonal carcinoma is a common ancestor of various non-seminomatous GCTs. See, Cheng et al. (2018) Nature Reviews Disease Primers (<https://doi.org/10.1038/s41572-018-0029-0>). Is there any reason that Authors want to emphasize this possibility?

Lines 421-423. The speculation, "... these results suggest that GCT cells retained the memory of the DMRT1-mediated epigenetic landscape in germline cells," is interesting but lacks sufficient basis. Authors introduced this speculation in their attempt to explain why the accessible chromatin in TGCTs (ATAC-seq data analysis) are enriched for DMRT1/6 motif whereas expression of DMRT1 is strongly suppressed in GCTs. It is understood that experimental evaluation of this speculation is beyond the scope of the current study; however, it does not seem important for the current study, either.

Lines 442-446. See comment on Lines 266-271.

Toshi Shioda, MD, PhD
Massachusetts General Hospital and Harvard Medical School

RE: NCOMMS-20-44315A

We thank the reviewers for their helpful and constructive comments and suggestions. We have performed the additional experiments to address their concerns. We believe that our manuscript has been substantially improved as a consequence. Particularly, we performed the ICR methylation analysis of S-OSKM tumors and found that, in sharp contrast to D-OSKM tumors, S-OSKM tumors exhibited increased levels of *H19* ICR methylation, which is consistent with our previous study demonstrating *in vivo* reprogramming-induced cancer cells with S-OSKM alleles exhibit increased DNA methylation at *H19* (Ohnishi K et al., *Cell* 2014). The results further strengthen our conclusion that D-OSKM tumor cells, but not S-OSKM tumor cells, have experienced the PGC-like reprogramming. We have responded to each point by the reviewers in the subsequent section. We hope that these experiments and our responses will clarify any concerns about the suitability of our manuscript for publication in *Nature Communications*.

Response to referee's comments:

Reviewer #1 (*in vivo* reprogramming) (Remarks to the Author):

Taguchi et al generated a new mouse model that allows in vivo reprogramming with high levels of OSKM. They found that in vivo reprogramming leads to the generation and propagation of PGC-like cells. In addition, reprogramming with higher levels of OSKM increases trophoctoderm differentiation within OSKM-derived tumors. They found Dmrt1, a transcription factor expressed in gonadal cells, as a promoter of in vivo reprogramming.

This is a nice piece of data that, on one hand, strengthens the concept of the extended potential of in vivo-generated PSC (already published but still unappreciated), and on the other hand provides new identities and regulators involved in in vivo reprogramming. Besides, the authors display an impressive amount of data, with state-of-the art technologies. Having said that, I would like to mention some concerns that may help the authors to improve the paper:

We thank reviewer #1 for positive comments.

*1. Authors suggest that in vivo reprogramming implies a linear sequence of events: differentiated cells \rightarrow PGC-like cells \rightarrow Trophoctodermal giant cells (TGCs). In other words, that PGC-like cells are an intermediate state, which have the potential to derive TGCs. Given that authors don't perform lineage tracing experiments with DAZL (or another marker of PGC-like cells) to demonstrate that TGCs are derived from PGCs, there is an alternative explanation: D-OSKM drives the generation of a mixture of independent cell populations: PGC-like cells (*Oct4+*/*DAZL+*), iPSC (*Oct4+*/*DAZL-*), and TGCs. Why do authors exclude this scenario, which in principle seems more plausible than the "strange" conversion (from a developmental point of view) of PGCs into TGCs? This alternative should be contemplated and discussed.*

We thank this this suggestion. We agree that lineage tracing experiments with DAZL could unequivocally demonstrate that PGC-like cells indeed give rise to TGCs. However, because D-OSKM system already harbors a Cre allele, this is not technically feasible in the current system. We would like to emphasize that D-iPSCs exhibit reduced methylation at imprinting control regions (ICRs). Given that D-OSKM tumor cells display reduced ICR methylation, which is one of PGC features, and that loss of ICR methylation could never be restored in embryonic lineage cells (Yagi M et al., *Nature*, 2017), these findings indicate that D-iPSCs have experienced a PGC-like state. Consistent with this, D-iPSCs exhibit increased accessibility at loci containing a DMRT1 motif and increased expression of *Dmrt1*, both observed in late PGCs. Additionally, we showed that D-iPSCs, even after a single cell-cloning, give rise to both TGCs and three germ layers. Taken together, these results support our

conclusion that the conversion occurs from PGC-like tumor cells to D-iPSCs, and finally to TGCs. This sequence is also supported by our lineage tracing experiment demonstrating that *Nanog*-expressing D-OSKM tumor cells give rise to GCTs *in vivo*.

2. In relation to the above point, the scheme in Fig 6j is very confusing. Why the arrow towards PSCs say *in vitro*? Why *DMRT1* is acting at the level of GCT-like cells and not at PSCs to induce Trophoblast differentiation?

We thank the reviewer for this suggestion. Based on this comment, we revised the scheme in Fig 6j. Now, the scheme describes the role of *DMRT1* in propagation of GCT-like cells.

3. It is not clear to me that D-OSKM induction drives the development of tumors that resemble germ cell tumors (GCT), and more specifically Type II non-seminomatous GCTs. Alternatively, D-OSKM tumors could be described as undifferentiated teratomas with a miscellanea of cells, some with PGC features, others with TGC features. It would be helpful to include some comparison (i.e. by transcriptomic profiles, markers) between ESC-derived teratomas, S-OSKM tumors, D-OSKM tumors and GCTs.

We thank the reviewer for this helpful comment. In the revised manuscript, we clarified the human tumor types, in relation to characteristics of D-OSKM tumors. Histological analyses showed that D-OSKM tumors resembled embryonal carcinomas (Fig. 1l). Considering that embryonal carcinoma is thought to be a common ancestor of other non-seminomatous GCTs, including GCTs containing extraembryonic lineage cells, the notion is also consistent with our findings that D-OSKM tumors contain TGCs and D-iPSCs harbor a potential to differentiate into TGCs. To further demonstrate that D-OSKM tumors resemble human GCTs, we compared gene expression profiles of D-OSKM tumors with those of human GCTs. Notably, D-OSKM tumors often exhibited increased expression of genes that are uniquely overexpressed in human GCTs (Fig. 1m). These results provide additional evidence that D-OSKM tumors have shared characteristics with human GCTs, especially embryonal carcinomas.

Figure 1l

Figure 1m

4. Page 14, line 352: Authors mention that D-iPSC are the only cells tested that have bidirectional potential, but in Fig 4c they show (in line with previous studies) that S-iPSC and ESCs also form tumors with TGCs. What is unique to D-iPSCs that is completely absent in S-iPSCs? The differences are quantitative, not qualitative.

In the revised manuscript, we additionally performed the ICR methylation analysis of S-OSKM tumors. Importantly, S-OSKM tumors exhibited increased levels of *H19* ICR methylation (Fig. 2e), which is consistent with our previous study demonstrating *in vivo* reprogramming-induced cancer cells with S-OSKM alleles exhibit increased

DNA methylation at *H19* (Ohnishi K et al., *Cell* 2014). Because D-OSKM tumors generally display a reduction in ICR methylation, the ICR methylation status provides qualitative differences. The difference also indicates that D-OSKM tumor cells, but not S-OSKM tumor cells, have experienced PGC-like reprogramming. We also confirmed that S-iPSCs display increased *H19* ICR methylation while D-iPSCs exhibit a reduction in *H19* ICR methylation (Fig. 4i).

Figure 2e

Figure 4i

5. Figure 2f,g: It would be informative to compare with the ICRs in S-OSKM tumors. Loss of imprinting could be a general phenomenon of in vivo reprogramming.

See above response to comment 4. Thanks to this reviewer's excellent suggestion, we were able to clarify qualitative differences between S-OSKM and D-OSKM tumors/iPSCs in the revised manuscript.

6. Following the same line, in Figure 4i it would be interesting to demonstrate the maintenance (or not) of ICR in S-iPSCs.

Based on this comment, we added ICR methylation analyses of S-iPSCs. We found that S-iPSCs exhibit increased methylation levels at *H19* ICR (Fig. 4i), in agreement with the findings that S-OSKM tumor cells show the hypermethylation at *H19* ICR.

7. EBs are normally the first differentiation step to generate PGCs in vitro. Surprisingly, in this favorable context, D-iPSCs do not activate any features of PGCs (Extended Data Figure 4h). Isn't this surprising? It seems that the PGC features are only generated in vivo.

Previous studies showed that PGCs can be reprogrammed to PSCs, which are referred to as embryonic germ cells (EGCs) under ex vivo condition (Matsui Y et al., *Cell* 1992, Resnick JL et al., *Nature* 1992). Importantly, EGCs share common features

with ESCs, but not germ cells. Thus, we think that a lack of PGC features is consistent with transcriptional features of EGCs. Accordingly, we discussed this in Results section. Additionally, although we agree that EBs could be the first differentiation step for PGC differentiation, EBs themselves do not exhibit PGC signatures.

8. *What happens if you monitor D-OSKM mice at longer time points? Do tumors become differentiated teratomas? This is an important point because GCTs do not become differentiated teratomas.*

Because most D-OSKM mice are highly morbid at the termination of the experiment (Dox ON 14 days+ OFF 7 days), we cannot monitor D-OSKM mice at longer point. In response to this reviewer' comment, we inoculated D-OSKM tumor cells into immunocompromised mice to form secondary tumors. We found propagation of OCT4+/NANOG+ cells, reminiscent of D-OSKM tumor cells, in secondary tumors (Extended Data Fig. 2h). These secondary tumors also contained the immature teratoma component. However, we would like to emphasize that embryonal carcinoma is thought to be a common ancestor of other non-seminomatous GCTs. Indeed, human embryonal carcinomas are often observed as mixed GCTs containing teratomas regions or extraembryonic cell regions. Therefore, histological findings in secondary tumors further support our conclusions that D-OSKM tumors resemble human embryonal carcinomas.

Extended Data Fig. 2h

9. *Page 16- Dmrt1 deficiency decreases in vivo reprogramming efficiency (Fig 6b,c), so it is not surprising that also decreases trophectoderm differentiation. Is Dmrt1 simply increasing in vivo reprogramming efficiency?*

Thank you for your comments. We think that DMRT1-mediated reprogramming is tightly related to the competence of trophectoderm differentiation. As this reviewer mentions, we think DMRT1 promotes *in vivo* reprogramming, which in turn elicits a trophectoderm differentiation potential.

10. *Page 19 Line 482: Authors state that D-iPSC exhibit a global transcriptional profile distinct from previously reported PSCs with similar differentiation potential. Where do they show these differences? Isn't this in conflict with the data shown in Extended Data Fig. 5b?*

In response to this reviewer' comment, we extracted upregulated/downregulated genes in D-iPSCs compared to S-iPSCs, then these genes were examined for their expression in previously reported PSCs with similar differentiation potential. In this comparison, we found no similarity with other PSC types (Extended Data Fig. 5c). Extended Data Fig. 5b (Extended Data Fig. 5d in the revised version) demonstrates that previously reported PSCs, as well as D-iPSCs, do not exhibit obvious similarity in global transcriptional signatures with developing embryos with totipotency,

suggesting that PSCs with extended differentiation potential *in vitro* are quite different from developing embryos with totipotency *in vivo*.

Extended Data Fig. 5c

Minor points:

11. The abstract is very confusing. I had to read the paper to understand the abstract.

According to the reviewer's comment, we revised the abstract.

12. Page 11, line 271: *Eomes* is also detectable in *in vitro* reprogramming.

Thank you for your comments. Based on the reviewer's comment, we toned down the description with regard to the *in vivo* specific induction.

13. Authors should demonstrate that they have efficiently abolished the expression of *CDX2* by CRISPR (as they do for *Tead4* KO in extended Fig 3D).

Because ESCs do not express *CDX2*, we were not able to perform Western blotting to confirm the lack of *CDX2* protein. Alternatively, we confirmed the lack of *CDX2* protein in somatic tissue by immunohistological analysis (Extended Data Fig. 3d, e).

14. Is it not clear why the pancreas and the kidney were chosen for reprogramming on the first place. Besides, kidney D-OSKM tumor and pancreas D-OSKM are used indistinctly with no clear criteria

In our previous studies, robust induction of OSKM transgene during *in vivo* reprogramming was observed in the kidney and pancreas. Consistently, the OSKM-induced mice exhibited overt phenotype in these organs (Ohnishi K et al., *Cell* 2014 and Shibata H et al., *Nat Commun.* 2018). Accordingly, we selected kidney and pancreas to compare phenotypes during S-OSKM and D-OSKM *in vivo* reprogramming. We mentioned it in Results section. We agree that some of experiments have been performed using one organ. However, we would like to emphasize that both kidney and pancreatic D-OSKM tumors exhibit similar histological/immunohistological features with similar kinetics after OSKM induction.

15. Page 9, lines 209-212: These sentences are not very clear. I don't understand the authors'

interpretation of sc-RNA-seq analysis. Please, clarify the message.

In scRNA-seq analysis, we detected pluripotency- and PCG-related genes only in a small subset of D-OSKM tumor cells, presumably because of low quality samples from *in vivo* tissues. Because the scRNA-seq data do not add important message in this manuscript, we decided to withdraw the data in the revised manuscript.

16. There are many mistakes in the numbering of the figures that difficult the reading and understanding of the manuscript (i.e. in line 217 Fig 2g should be 2f, in line 219 Fig 2h,i should be Fig 2 g,h, in line 221 Fig 2h should be 2g, in line 366, Fig 6d should be extended Fig 6d, line 403 Fig 6d shouldn't be 6b, 6h?).

We sincerely apologize for the mistakes. We have corrected the numbering of the figures.

17. Page 16, line 403: I guess authors wanted to say “However, Dmrt1-KO D-OSKM tumors still contained clusters of TGCs”, and the referenced figure should be 6h.

According to the comments, we have revised the manuscript.

18. Page 9, line 203-204; Page 12 line 284: Some references should be added here.

Thank you for your comments. In the revised manuscript, we added references.

19. Line 230 is missing the abbreviation of trophoblast giant cells (TGCs).

Thank you for your comments. We added the abbreviation.

Reviewer #2 (ATAC-seq, scRNA-seq) (Remarks to the Author):

In this study, the authors found that *in vivo* expression of OSKM caused increased expression of PGC-related genes and provoked the genome-wide ICR demethylation in adult somatic cells. RNA-seq and methylation sequencing also confirmed this discovery. This reprogramming drove the development of a subgroup of human GCT-like cancers that exhibited trophoblast differentiation. Furthermore, the authors identified DMRT1 participated in reprogramming and associated with trophoblast competence using ATAC-seq technology. In general, this work is well organized and systematic, serving as a good reference for implications for GCT development and the acquisition of totipotency-like features by somatic cells. The sequencing data analysis is weak. I have a few comments:

We thank reviewer #2 for positive comments.

1. In Fig 5a, ATAC-seq results indicated that DMRT1 was involved in reprogramming, however there were only 270 D-iPSC-specific peaks. The authors should change the method to identify sample type-specific peaks, because in Fig 1e, RNA-seq results suggested that there were great differences between D-OSKM and S-OSKM in gene expression. The authors should provide basic information about the ATAC-seq data, such as: how many replicates were used? how was the data quality? how many peaks in total were identified? How the differential peaks were evaluated? and the thresholds were used to indicate the level of significance. Without this information, it is difficult convince the readers.

Thank you for your comments. ATAC-seq was conducted using PSCs (ESCs, S-iPSCs and D-iPSCs) where only a modest transcriptional difference was detected, which may explain the small number of differential peaks in ATAC-seq analysis. In the revised manuscript, we have provided basic information about the ATAC-seq data, which include the total read number of sequencing reads and the mapping efficiency (Extended Data Fig. 6a). Also, we have provided the details of the number of common/differential peaks in each comparison, which was used to identify overlapping peaks (Extended Data Fig. 6a). Furthermore, we performed ATAC-qPCR at the representative peaks to confirm the results in ATAC-seq using multiple clones (Fig. 5b and Extended Data Fig. 6c).

Extended Data Fig. 6a

a

	Total read number	Mapping efficiency (%)
ESCs (LacZ)	125,787,010	96.7
S-iPSC (p3)	117,389,850	94.5
D-iPSC (p3)	116,012,340	96.4
D-iPSC (p25)	141,428,232	96.3

	Common peaks	Differential peaks	
		Enriched in D-iPSC (p3)	Depleted in D-iPSC (p3)
vs ESCs (LacZ)	55,911	2109	6608
vs S-iPSC (p3)	56,078	1860	6623
vs D-iPSC (p25)	55,458	1787	3773
	Overlapping peaks 270	Overlapping peaks 992	

Figure 5b

The method used to calculate the motif enrichment score in Fig 7d does not consider the number of TF binding sites in the whole genome, and therefore is problematic.

We thank the reviewer for important comments. To access the genome-wide distribution of TF binding sites as background controls, we focused on open chromatin regions which are identified by ATAC-seq. To this end, we examined the motif enrichment score at pan-cancer peaks (562,709 reproducible ATAC-seq peaks

in 410 samples from various tumor types) (Ryan Corces M et al., *Science* 2018), which should cover most general open chromatin regions. Importantly, we confirmed that DMRT1/6 motif is significantly enriched in TGCTs when compared to the background distribution (Fig. 7b and Extended Data Fig.9c,d).

Figure 7b

Extended Data Fig. 9d

2. The authors claimed that they found a small subset of D-OSKM tumor cells through single-cell RNA-seq (Fig 2d) and showed the expressions of several PGC-related genes (Extended Data Fig 2b), however it seems that none of the D-OSKM tumors subtype cells expressed these four genes except for PGC cell subpopulation in Extended Data Fig 2b. Which group is the small subset of D-OSKM tumor cells that author claimed? In addition to these four genes, what is the overall expression level of PGC-related genes in these tumors? How many cell subtypes were found through scRNA-seq, and what are they? Is cell clustering in Fig 2e over fitting, since there are many cell clusters contain very few number of cells? What are the parameters used to identify these cell subtypes, and how do the authors make sure the clustering is right? Does it mean that even though D-OSKM tumors express PGC-related genes, they were still different in transcriptome level? Fig. 2e and Extended Data Fig 2b should be combined and better labeling should be provided, otherwise readers won't understand which cell type is which.

We appreciate this important comment. As this reviewer pointed out, we agree that our scRNA-seq analysis did not detect clear subpopulation that express multiple PGC-related genes simultaneously. Moreover, we detected pluripotency-related gene expression only in a small subset of D-OSKM tumor cells, presumably because of low quality samples obtained from *in vivo* tissues. Because the scRNA-seq data do not add important message in this manuscript, we decided to withdraw the data in the revised manuscript.

3. In line 276, the authors described "lack the evidence of differentiation into three germ layers". Why not test the capacity to form teratomas of the minced D-OSKM kidney tumors directly?

We thank the reviewer for helpful comments. In response to this reviewer' comment, we inoculated D-OSKM tumor cells into immunocompromised mice to form secondary tumors. We found propagation of OCT4+/NANOG+ cells, reminiscent of D-OSKM tumor cells, in secondary tumors (Extended Data Fig. 2h). However, these secondary tumors also contained the immature teratoma component, which suggest that these cells could acquire differentiation potential into three germ layers after inoculation. Accordingly, we have toned down the statement and removed the description.

Extended Data Fig. 2h

4. The authors mentioned that *DUX* was responsible for the unique transcriptional signature in the 2C-like state, but in this model, the expression of *DUX* is very low. So, what is the difference between these two pathways? The author should compare the transcriptome difference between the two pathways.

Based on this reviewer' comment, in the revised manuscript, we showed expression levels of *Duxf3* and *Zscan4d*, demonstrating no upregulation in D-iPSCs (Extended Data Fig. 5b). We also extracted upregulated/downregulated genes in D-iPSCs compared to S-iPSCs, then these genes were examined for their expression in previously reported PSCs with similar differentiation potential. Again, we failed to find similarity in transcriptional signatures with other PSC types in this comparison (Extended Data Fig. 5c). Although we were not able to identify the specific pathway that characterize D-iPSC properties, we found that *Dmrt1* is commonly overexpressed in all PSC types with expanded differentiation potential (Extended Data Fig. 10d). We would like to examine the role of the increased *Dmrt1* in these PSC types in future experiments.

Extended Data Fig. 5b

Extended Data Fig. 5c

5. The authors mentioned that D-OSKM tumors resembled human GCTs in the article, but the authors need to compare their differences at transcriptome level to prove D-OSKM tumors can be used as a model for GCT development.

Thank you for helpful comments. In response to this comment, we compared transcription signatures of D-OSKM tumors with those of human GCTs. Notably, D-OSKM tumors often exhibited increased expression of genes that are uniquely overexpressed in human GCTs (Fig. 1m). These results provide additional evidence that D-OSKM tumors have shared characteristics with human GCTs.

Figure 1m

6. The color bar of Fig 7a and Extended Data Fig 5b have no color gradient.

We apologize for the lack of color gradient. It seems that the color gradient has disappeared during the PDF conversion. We have corrected it.

7. In Fig 2f and Fig 4i, the authors need to provide quantitative values to measure the level of methylation in different samples. No description about the coordinates will make reader very confused.

We have added the percentage of methylated CG sites in total CG sites analyzed in Fig. 2e and Fig. 4i. In addition, we added ICR methylation data for S-OSKM tumors and S-iPSCs, which underscores differences between S-OSKM and D-OSKM *in vivo* reprogramming.

8. In lines 212 and 217, the figure label is incorrect, "However, these tumor cells were identified as a distinct population from PGCs and ESCs (Fig. 2e, f)", it should be "However, these tumor cells were identified as a distinct population from PGCs and ESCs (Fig. 2e)"

We sincerely apologize for the mistake. We have corrected the numbering of the figure.

9. The author showed induction of OSKM in the kidney and pancreas drove *in vivo* reprogramming and propagation of GCT-like tumor cells, why not try to induction of OSKM in germ cells?

We thank this reviewer for this important comment. We agree induction of OSKM in germ cells could directly demonstrate the involvement of reprogramming in GCT development. However, it is technically difficult because our tet-On system does not allow robust induction in germ cells (Beard C et al., *Genesis*, 2006). We would like to perform the suggested experiments with different strategy in future experiments.

Reviewer #3 (germ cells, germ cell tumours) (Remarks to the Author):

The study described in this manuscript is a straightforward and significant extension of a series of preceding research performed by the group of Authors on in vivo carcinogenesis of mouse models that transiently express the OSKM reprogramming factors under control of exposure to doxycycline. In the current study, tissue-specific, doxycycline-inducible expression of large amounts of the OSKM reprogramming transcription factors in mouse kidney or pancreas in vivo resulted in generation of highly malignant cells (D-OSKM tumor cells), which express some of the pluripotency marker genes and show several signs of totipotency. Authors provided a series of persuasive evidence that DMRT1 plays critical roles in the observed gain of totipotency.

I think that the presented research has significant scientific merits, and it may be worth publishing from Nature Communications, pending revisions addressing the following concerns:

We thank reviewer #3 for positive comments.

Number 1: Authors argue that INCREASED LEVELS of OSKM expression is the cause of the phenomenon. However, their own data indicate that stoichiometry of expression of the four factors is significantly different in the D-OSKM MEFs from that observed with the previously reported, S-OSKM MEFs. Authors are requested to address possible contributions of the IMBALANCED expression of the OSKM factors to the observed phenomena.

Thank you for your thoughtful comments. We agree that stoichiometry of reprogramming factors has a significant impact on somatic cell reprogramming. Accordingly, we performed semi-quantification of reprogramming factor proteins in Western blotting of D-OSKM MEFs. We have included these data in the Figure legend (Fig. 1c). Indeed, we observed a subtle alteration in stoichiometry (relative increase of OCT4 protein). However, a previous *in vitro* study demonstrated that increased OCT4 dose not significantly enhance OSKM reprogramming in MEFs (Kim S et al., *Stem Cell Reports*, 2015, Hammachi F et al., *Cell Reports*, 2012). In addition, the decreased, but not increased, expression of OCT4 has been implicated in trophectoderm differentiation in ESCs (Niwa H et al., *Nature Genetics*, 2000). Collectively, although we cannot exclude the possibility that this difference has some effects, we don't think it has significant impact on the unique phenotype of D-OSKM-mediated reprogramming.

*Number 2: Reproducibility of the experiments GENERATING the D-OSKM ESCs and MEFs is not clearly documented. How many independent D-OSKM ESC clones were isolated in the current study? Do they show the same, or at least similar, phenotypes in vitro and in vivo? It is also important to describe how the individual D-OSKM ESC clones were characterized after electroporation of the linearized target vector in sufficient details. I am assuming that both S-OSKM and D-OSKM ESCs were genetically engineered by using the targeted knock-in technique at the *Col1a1* and *Rosa26* loci, but procedures of screenings and/or characterization is not described in the manuscript. Since electroporation of linearized plasmids may also generate insertions of plasmid concatemers at non-targeted sites in the genomic DNA, evaluation of such off-target insertions is often critical for studies on carcinogenesis using cell culture models.*

We thank the reviewer for this comment. We have used a single ESC line for each genotype to induce S-OSKM and D-OSKM both *in vitro* and *in vivo*. However, we would like to emphasize that both kidney and pancreatic D-OSKM tumors exhibited similar histological/immunohistological features with similar kinetics after OSKM induction. Importantly, these experiments (in the pancreas and kidney) were conducted using different ESC clones, demonstrating that the unique D-OSKM phenotypes are not attributable to the ESC-clonal difference. In this study, ESC

clones for OSKM induction were generated by gene-targeting methods. We described the targeting methods, including Flip-in mediated targeting in Methods section. We confirmed the correct targeting by PCR-based genotyping. To further confirm that the observed findings in D-OSKM mice could indeed be attributed to higher levels of OSKM expression, we decreased the dose of Dox during *in vivo* D-OSKM reprogramming. Treatment with a lower dose of Dox led to mature teratoma formation in both the kidney and pancreas of D-OSKM mice, which, we believe, excludes the possibility of the ESC-clone-specific phenotypes.

Number 3: Since the D-OSKM is a mouse model, I am a little bit surprised by the total lack of mentions to mouse embryonal carcinoma cell culture models in the manuscript whereas Authors attempt to link the biological characteristics of the D-OSKM tumor cells to human embryonal carcinoma. The phenomenon that mouse embryonal carcinoma cells, which are highly malignant, are capable of contributing to normal tissue development in mice when injected into mouse embryos is already known (e.g., See, Astigiano et al. 2005, "Fate of embryonal carcinoma cells injected into postimplantation mouse embryos" Differentiation 73:484-490). Roles of Dmrt1 in teratoma incidence in 129 mice as a repressor of pluripotency are also documented (See, a review article of Bustamante-Marin, Garness, and Capel 2013, "Testicular teratomas: an intersection of pluripotency, differentiation, and cancer biology" Int. J. Dev. Biol 57:201-210). To me, the D-OSKM system seems like an excellent model of MOUSE embryonal carcinomas, and its relevance to human embryonal carcinoma sounds indirect. Authors are requested to address why they think their model is more closely linked to human embryonal carcinoma than the same type of tumors or tumor cells in mice. In addition, I urge Authors to specify the human tumor types a little bit more carefully. It would be fair to suggest the resemblance of their D-OSKM model to human embryonal carcinomas based on their presented data, but I see multiple occasions that they attempted to extend their claim to Type II GCTs (including both seminomatous and non-seminomatous tumors) or even generally GCTs.

Thank you so much for helpful comments. We appreciate these suggestions regarding human GCT types. We agree that D-OSKM tumor cells resemble human embryonal carcinomas, but not seminomatous tumors. Accordingly, we have revised our manuscript. We also mentioned a previous study with a mouse embryonal carcinoma cell culture model, which demonstrated that mouse embryonal carcinoma cells (teratocarcinoma cells, which were derived by grafting early mouse embryos into the testes and maintained as an ascites tumor) are capable of contributing to normal tissue development in mice when injected into mouse embryos (Discussion section). We would like to emphasize that D-OSKM tumor cells exhibit ICR demethylation, which is unique epigenetic alterations in human GCTs. We also showed histological similarity of D-OSKM tumors with human embryonal carcinomas (Fig. 1l). In the revised manuscript, we compared gene expression profiles of D-OSKM tumors with those of human GCTs. Notably, D-OSKM tumors often exhibited increased expression of genes that are uniquely overexpressed in human GCTs (Fig. 1m). Taken together, these results further support our conclusion that D-OSKM tumors have shared characteristics with human GCTs.

Figure 1l

Figure 1m

[Specific Comments]

Lines 137-140. The sentence, “The results indicate that different levels of OSKM expression resulted in a substantial difference in global transcriptional signatures, which presumably contributed to the differences in the efficiency of invitro iPSC derivation following withdrawal of OSKM expression,” has the following three concerns:

Number 1: Authors stated in Lines 74-76, “Notably, the details of the reprogramming process depends on cell-intrinsic factors such as cell type, as well as the LEVELS AND STOICHIOMETRIES of reprogramming factors.” Figure 1b (RT-qPCR) shows that mRNA expression of the reprogramming factors differs remarkably between the S-OSKM and D-OSKM MEFs in not only their LEVELS but also STOICHIOMETRIES. However, the sentence of Lines 137-140 mentions only the different LEVELS. Authors are requested to explain why they think that only the increased levels – not their apparent imbalance – of the reprogramming factor expression affected the transcriptomes.

Please see above response to the comment Number 1.

Number 2: The basis of the claim that the observed substantial differences in global transcriptional signature likely contribute to the reprogramming efficiency needs more explanations. Because the PCA plot (Figure 1e) does not tell the degree of transcriptomal impact (number of affected genes of magnitudes of changes in expression), the claimed “substantial differences in global transcriptional signature” is not easily evaluated. Even if expression of a number of genes differ significantly between the S- and D-OSKM MEFs, whether such differential expression affects the reprogramming efficiency would need further explanations.

As this reviewer pointed out, we agree that we have not shown “substantial differences in global transcriptional signature”. Accordingly, we toned down the description. In the revised manuscript, we demonstrated that D-OSKM MEFs exhibit rapid repression of MEF-related genes (Extended Data Fig 1c). Given that the repression of somatic cell transcriptional signatures plays a crucial role in initial stage of successful reprogramming (Chronis C, Cell 2017), the rapid repression may be associated with increased reprogramming efficiency in D-OSKM MEFs. We also performed Gene ontology enrichment analysis for upregulated/downregulated genes in D-OSKM MEFs (Day 14) and demonstrated that keratinocyte-related genes and immune response-related genes are upregulated and downregulated, respectively. We have included these results in Extended Data Fig 1d.

Extended Data Fig. 1c, 1d

Number 3: In Figures 1b, 1d, and 1e, I can see three dots for each bar. Are these dots represent three independent clones of S- or D-OSKM? Alternatively, are these data show outcomes of three independent experiments performed using only a single clone of S- or D-OSKM? Authors are requested to clarify the number of independent S/D-OSKM clones involved in these figures and discuss the experimental reproducibility.

As mentioned above, we have used only a single ESC line for each genotype to induce S-OSKM and D-OSKM. Regarding the experimental reproducibility of different ESC clones, please see response to the comment Number 2.

Lines 161-164. The sentence claims that D-OSKM reprogrammable mice developed tumors in kidney and pancreas with nearly complete penetrance. Authors are requested to present numbers of mice and tumor incidence for S- and D-OSKM mice.

We thank this reviewer for this important comment. We have provided the number of tumor incidence in the revised manuscript. We think that absence of tumors in a small number of D-OSKM mice is attributed to the lack of chimeric contribution of the target organ. In Fig. 1i and 1n, we added two S-OSKM tumors (total 18 tumors) and three D-OSKM tumors (total 20 tumors) for the analysis of OCT4-positive cell area in tumors.

Lines 185 and 189. See, comment on Lines 137-140, Number 1.

Please see above response to the comment Number 1.

Lines 191-193. The claim, “We found that D-OSKM tumors consisted of OCT4-positive proliferating cells [that] also expressed NANOG, INDICATING that D-OSKM tumors RESEMBLED human GCTs,” sounds too strong based solely on mere expression of two pluripotency marker genes. Note that further evidence linking the D-OSKM tumor model to human embryonal carcinoma cells are described later in the manuscript. It is acceptable to state here, “We found that D-OSKM tumors consisted of OCT4-positive proliferating cells also expressed NANOG. Expression of these pluripotency markers is also known as characteristics of several subtypes of human germ cell tumors such as embryonic carcinomas or seminomas.”

Thank you for your helpful suggestions. According to the suggestion, we have revised the description.

Lines 193-199. The presented pathological characteristics of D-OSKM tumors (invasive growth and central necrosis) are common features of highly malignant tumors. Since these are not specific characteristics of embryonic carcinoma, the last sentence, “Together, these findings show that ... resulted in development of cancers that shared characteristics with human malignant type II non-seminomatous GCTs,” sounds too strong. It is acceptable to state, “... resulted in development of cancers with strongly malignant characteristics.”

Thank you for your helpful comments. According to the suggestion, we have revised the description.

Lines 212, 217, and 219. I think the reference to figures in these lines are incorrect. Line 212: (Fig. 2e,f) -> (Fig. 2e). Line 217: (Fig. 2g) -> (Fig. 2f). Line 219: (Fig. 2h,I) -> (Fig. 2g, h). Note that there is no Figure 2I.

We sincerely apologize for the mistake. We have corrected the numbering of the figure.

Lines 213-224, Fig. 2f,g,h, Extended Data Fig. 2c,d. The observed ICR-specific strong demethylation in the D-OSKM tumors is interesting. If tumor RNA samples are still available, Authors should examine mRNA expression of the imprinting genes whose ICR is strongly demethylated. Since Authors have RNA-seq data of S- and D-OSKM MEFs in the course of in vitro reprogramming towards PSCs (Fig. 1e), it would be appreciated if Authors examine expression of the imprinting gene mRNA transcripts.

We thank this reviewer for the thoughtful comments. Because expression levels of imprinting genes significantly alter depending on cellular context, it is important to use the identical cell type for the comparison. Therefore, we compared *H19* expression levels in PSCs. We found that *H19* is upregulated in D-iPSCs when compared with S-iPSCs or ESCs (Extended Data Fig. 5a), which is consistent with reduced DNA methylation levels at the *H19* DMR in D-iPSCs.

Extended Data Fig. 5a

Lines 241-245. Authors claim that the presence of cells reminiscent of trophoblastic giant cells and their precursors in the D-OSKM tumors provides additional evidence that D-OSKM tumors share properties with human GCTs. I would say that the D-OSKM tumors show signs of totipotency, which is a shared feature with human embryonal carcinoma.

We appreciate the helpful comments. According to the reviewer's comments, we revised the description throughout the manuscript.

Lines 246-252. The presented lineage tracing data demonstrate that Nanog-expressing cells give rise to cells resembling the trophoblast giant cells in the S-OSKM tumors. Although technical limitations may prevent lineage tracing experiments directly using the D-OSKM tumors, Authors are requested to carefully discuss whether the data obtained from the S-OSKM model can be adequately extrapolated to biology of the D-OSKM cells.

We appreciate the helpful comments. Based on the reviewer's comments, we toned down the description in the revised manuscript.

Lines 260-261 and Fig. 3i, j, Extended Data Fig. 3e. Areas of TGC clusters should be evaluated using PL-1 immunohistochemistry. The text referring to Fig. 3c (Lines 229-232) suggest that area of TGC cluster was evaluated by PL-1 immunohistochemistry, but neither figure legend nor Method (Lines 782-797) clearly states so. H&E staining images presented as Fig. 3i are not sufficiently informative to identify areas occupied by TGC clusters. If PL-1 staining was not involved in calculating areas of TGC clusters presented in either Fig. 3c or Fig. 3j, strong justification for the use of plain H&E images is required. This comment is also applicable to Fig. 4c, Extended Data Fig. 4e, and Fig. 5i.

Thank you for your comments. A previous study demonstrated that there exist diverse subtypes of TGCs, including PL-1-negative TGCs in the mouse placenta (Simmons D et al., *Developmental Biology*, 2007). Consistent with this, we observed PL-1-negative giant cells in D-OSKM tumors, which are recognized as TGCs on HE stained sections. Therefore, we used plain HE sections to identify TGCs and TGC clusters.

Lines 266-271. Authors observed the increased expression of PGC- and trophoblast-related genes only during the in vivo reprogramming process. Authors are requested to discuss whether the in vitro cell culture conditions during reprogramming are adequate to support survival of PGC-like or TGC-like cells derived from the D-OSKM ESCs. Even if such cells could actually be generated in both the in vitro and in vivo reprogramming conditions, these cells may not be able to survive and/or proliferate in the cell culture conditions optimized for reprogramming.

We thank the reviewer for thoughtful comments. We agree it is possible that PGC-like or TGC-like cells indeed appear during *in vitro* D-OSKM reprogramming but cannot be maintained under *in vitro* culture condition. Although we did not observe the increased apoptotic cells (measured by FACS with Annexin V staining, see below) in D-OSKM induced MEFs, we cannot completely exclude the possibility. We changed the description as follows; “longer periods of OSKM expression, along with features of the *in vivo* environment, are needed for the expansion of cells with PGC-associated signatures”.

Line 366. I think the referenced figure (Fig. 6d) is incorrect. It should be (Extended Data Fig. 6d).

We sincerely apologize for the mistake. We have corrected the numbering of the figure.

Line 403. Authors claim that D-OSKM tumors still contained clusters of trophoblast giant cells (Fig. 6b), but is hard to identify the claimed trophoblastic giant cells in the image presented as Fig. 6b.

We showed the higher magnification of the histological image of *Dmrt1* KO D-OSKM tumors containing TGCs (Extended Data Fig. 8e).

Line 411. Authors state that seminomas may represent precursors of non-seminomas. Although this possibility has been proposed, it is not well-demonstrated yet. Currently accepted notion is that both seminomas and embryonal carcinomas are derived directly from GCNIS and that embryonal carcinoma is a common ancestor of various non-seminomatous GCTs. See, Cheng et al. (2018) Nature Reviews Disease Primers (<https://doi.org/10.1038/s41572-018-0029-0>). Is there any reason that Authors want to emphasize this possibility?

We appreciate the helpful comments. According to the reviewer's comments, we have revised the manuscript. Now we propose that D-OSKM tumors resemble human embryonal carcinomas, which could progress into GCTs containing trophoblasts.

Lines 421-423. The speculation, "... these results suggest that GCT cells retained the memory of the DMRT1-mediated epigenetic landscape in germline cells," is interesting but lacks sufficient basis. Authors introduced this speculation in their attempt to explain why the accessible chromatin in TGCTs (ATAC-seq data analysis) are enriched for DMRT1/6 motif whereas expression of DMRT1 is strongly suppressed in GCTs. It is understood that experimental evaluation of this speculation is beyond the scope of the current study; however, it does not seem important for the current study, either.

Thank you for the comment. Because GCNIS expresses higher levels of DMRT1 and human GCTs harbor increased accessibility of loci containing DMRT1 motif, we would like to propose the possibility that DMRT1-mediated reprogramming may be involved in the progression of GCNIS into embryonal carcinomas, which further progress into GCTs containing trophoblasts. Still, we agree that we do not provide sufficient evidence to establish the model, therefore, in response to the reviewer's comments, we have toned down our statement and removed Fig. 7h.

Lines 442-446. See comment on Lines 266-271.

Please see comments above.

Reviewers' Comments:

Reviewer #1:

Remarks to the Author:

The authors have made a great work and the manuscript has greatly improved in this revised version. In particular, I appreciated their efforts in performing the ICR methylation analysis in S-OSKM tumors. The authors have satisfied most of my concerns, and I feel very comfortable suggesting this manuscript for publication in Nature Communications, pending some minor changes/discussion that I think they will improve the final manuscript:

-Regarding my comment: "Authors suggest that in vivo reprogramming implies a linear sequence of events: differentiated cells>PGC-like cells>Trophoectodermal giant cells (TGCs). In other words, that PGC-like cells are an intermediate state, which have the potential to derive TGCs. Given that authors don't perform lineage tracing experiments with DAZL (or another marker of PGC-like cells) to demonstrate that TGCs are derived from PGCs, there is an alternative explanation: D-OSKM drives the generation of a mixture of independent cell populations: PGC-like cells (Oct4+/DAZL+), iPSC (Oct4+/DAZL-), and TGCs. Why do authors exclude this scenario, which in principle seems more plausible than the "strange" conversion (from a developmental point of view) of PGCs into TGCs? This alternative should be contemplated and discussed".

Authors do not unequivocally demonstrate that PGC-like cells indeed give rise to TGCs. I do not think lineage-tracing experiments are needed at this point, but the possibility that we suggested could be included in the discussion.

- In Figure 1l it would be helpful to have an histology image of human embryonal carcinomas to be able to compare them with the D-OSKM tumors.

- Extended Data Fig. 2h. Do these secondary tumors contain tissues belonging to the three embryonic layers? Or are they only immature?

-The abstract is more clear now. However, the sentence 57-59 is still confusing. Who contributes to adult somatic cells? iPSCs or trophoblasts?

- The abbreviation of trophoblast giant cells is still missing in the main text in page 9 line 220 in the new manuscript

Reviewer #2:

Remarks to the Author:

My concerns have been nicely addressed and I recommend for publication.

Reviewer #3:

Remarks to the Author:

I appreciate Authors for addressing all my concerns. I do not have any further comments.

Toshi Shioda, MD, PhD
MGH Center for Cancer Research & Harvard Medical School

RE: NCOMMS-20-44315A

We thank the reviewer #1 for his/her helpful and constructive comments and suggestions. Based on these comments, we have revised our manuscript. We have responded to each point by the reviewer #1 in the subsequent section. We hope that our responses will clarify remaining concerns about the suitability of our manuscript for publication in *Nature Communications*.

Response to referee's comments:

Reviewer #1 (Remarks to the Author):

*The authors have made a great work and the manuscript has greatly improved in this revised version. In particular, I appreciated their efforts in performing the ICR methylation analysis in S-OSKM tumors. The authors have satisfied most of my concerns, and I feel very comfortable suggesting this manuscript for publication in *Nature Communications*, pending some minor changes/discussion that I think they will improve the final manuscript:*

-Regarding my comment: "Authors suggest that in vivo reprogramming implies a linear sequence of events: differentiated cells>PGC-like cells>Trophoectodermal giant cells (TGCs). In other words, that PGC-like cells are an intermediate state, which have the potential to derive TGCs. Given that authors don't perform lineage tracing experiments with DAZL (or another marker of PGC-like cells) to demonstrate that TGCs are derived from PGCs, there is an alternative explanation: D-OSKM drives the generation of a mixture of independent cell populations: PGC-like cells (Oct4+/DAZL+), iPSC (Oct4+/DAZL-), and TGCs. Why do authors exclude this scenario, which in principle seems more plausible than the "strange" conversion (from a developmental point of view) of PGCs into TGCs? This alternative should be contemplated and discussed".

Authors do not unequivocally demonstrate that PGC-like cells indeed give rise to TGCs. I do not think lineage-tracing experiments are needed at this point, but the possibility that we suggested could be included in the discussion.

We thank the reviewer for this comment. Based on the unique dynamics of ICR methylation: DNA methylation status at both methylated and unmethylated alleles is stably maintained in somatic cells, we consider that D-OSKM cells experienced PGC-related reprogramming differentiated into trophoblasts in our experimental settings. However, we agree that the lineage tracing experiment with a PGC-specific reporter should provide unequivocal evidence for this conversion. Accordingly, we included the description in the discussion.

- In Figure 11 it would be helpful to have an histology image of human embryonal carcinomas to be able to compare them with the D-OSKM tumors.

We thank the reviewer for this comment. In the revised manuscript, we included histological images of human embryonal carcinomas in Supplementary Fig. 2h.

h

- *Extended Data Fig. 2h. Do these secondary tumors contain tissues belonging to the three embryonic layers? Or are they only immature?*

These secondary tumors contained the immature teratoma component with evidence of differentiation into the three embryonic germ layers. This is consisted with the fact that immature teratomas exhibit differentiation into three germ layers and that human embryonal carcinomas are often developed as mixed GCTs containing teratoma regions or extraembryonic cell regions.

- *The abstract is more clear now. However, the sentence 57-59 is still confusing. Who contributes to adult somatic cells? iPSCs or throphoblasts?*

In response to this reviewer' comment, we modified the abstract as follows: Moreover, these tumor cells give rise to induced pluripotent stem cells (iPSCs) with expanded differentiation potential into trophoblasts. Remarkably, the tumor-derived iPSCs are able to contribute to non-neoplastic somatic cells in adult mice.

- *The abbreviation of trophoblast giant cells is still missing in the main text in page 9 line 220 in the new manuscript*

In response to this reviewer's comment, we added the abbreviation of trophoblast giant cells in page 9.

Reviewer #2 (Remarks to the Author):

My concerns have been nicely addressed and I recommend for publication.

We thank the reviewer #2 again for his/her helpful and constructive comments.

Reviewer #3 (Remarks to the Author):

I appreciate Authors for addressing all my concerns. I do not have any further comments.

We thank the reviewer #3 again for his/her helpful and constructive comments.